# RETHINKING UNIFORMITY IN SELF-SUPERVISED REPRESENTATION LEARNING

## ABSTRACT

Self-supervised representation learning has achieved great success in many machine learning tasks. Many research efforts tends to learn better representations by preventing the model from the *collapse* problem. Wang & Isola (2020) open a new perspective by introducing a uniformity metric to measure collapse degrees of representations. However, we theoretically and empirically demonstrate this metric is insensitive to the *dimensional collapse*. Inspired by the finding that *representation that obeys zero-mean isotropic Gaussian distribution is with the ideal uniformity*, we propose to use the Wasserstein distance between the distribution of learned representations and its ideal distribution with maximum uniformity as a quantifiable metric of *uniformity*. To analyze the capacity on capturing sensitivity to the dimensional collapse, we design five desirable constraints for ideal uniformity metrics, based on which we find that the proposed uniformity metric satisfies all constraints while the existing one does not. Synthetic experiments also demonstrate that the proposed uniformity metric is capable to distinguish different dimensional collapse degrees while the existing one in (Wang & Isola, 2020) is insensitive. Finally, we impose the proposed *uniformity* metric as an auxiliary loss term for various existing self-supervised methods, which consistently improves the downstream performance.

## 1 INTRODUCTION

Self-supervised representation learning has become increasingly popular in machine learning community (Chen et al., 2020; He et al., 2020; Caron et al., 2020; Grill et al., 2020; Chen & He, 2021; Zbontar et al., 2021), and achieved impressive results in various tasks such as object detection, segmentation, and text classification (Xie et al., 2021; Wang et al., 2021b; Yang et al., 2021; Zhao et al., 2021; Wang et al., 2021a; Gunel et al., 2021). Aiming to learn representations that are invariant under different augmentations, a common practice of self-supervised learning is to maximize the similarity of representations obtained from different augmented versions of a sample by using a Siamese network (Bromley et al., 1994; Hadsell et al., 2006). However, a common issue with this approach is the existence of trivial constant solutions that all representations collapse to a constant point (Chen & He, 2021), as visualized in Fig. 1, known as the *collapse* problem (Jing et al., 2022).

Many efforts have been made to prevent the vanilla Siamese network from the collapse problem. The well-known solutions can be summarized into three types: contrastive learning (Chen et al., 2020; He et al., 2020; Caron et al., 2020), asymmetric model architecture (Grill et al., 2020; Chen & He, 2021), and redundancy reduction (Zbontar et al., 2021; Zhang et al., 2022b). While these solutions could avoid the complete constant collapse, they might still suffer from a *dimensional collapse* (Hua et al., 2021) in which representations occupy a lower-dimensional subspace instead of the entire available embedding space (Jing et al., 2022), as depicted in the Fig. 1. Therefore, to show the effectiveness of the aforementioned approaches, we need a quantifiable metric to measure the collapse degree of learned representations.

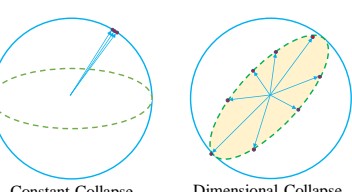

Constant Collapse Dimensional Collapse

Figure 1: The left figure presents constant collapse, and the right figure visualizes dimensional collapse.

To gain a quantifiable analysis of collapse degree, recent works (Arora et al., 2019; Wang & Isola, 2020) propose to divide the loss function into *alignment* and *uniformity* terms. For instance, recent objective functions such as InfoNCE (van den Oord et al., 2018) and cross-correlation employed in Barlow Twins (Zbontar et al., 2021) could be divided into two terms. These *uniformity* terms could explain the degree of collapse to some extent, since they measure the variability of learned representations (Zbontar et al., 2021). However, the calculation of these *uniformity* terms relies on the choice of anchor-positive pair, making them hard to be used as general metrics. Wang et al (Wang & Isola, 2020) further propose a formal definition of *uniformity* metric via Radial Basis Function (RBF) kernel (Cohn & Kumar, 2007). Despite its usefulness (Gao et al., 2021; Zhou et al., 2022), we theoretically and empirically demonstrate that the metric is insensitive to the dimensional collapse.

In this paper, we focus on designing a new uniformity metric that could capture salient sensitivity to the dimensional collapse. Towards this end, we firstly introduce an interesting finding that *representation that obeys zero-mean isotropic Gaussian distribution is with the ideal uniformity*. Based on this finding, we use the Wasserstein distance between the distribution of learned representations and the ideal distribution as the metric of *uniformity*. By checking on five well-designed desirable properties (called 'desiderata') of *uniformity*, we theoretically demonstrate the proposed uniformity metric satisfies all desiderata while the existing one (Wang & Isola, 2020) does not. Synthetic experiments also demonstrate the proposed uniformity metric is capable to quantitatively distinguish various dimensional collapse degrees while the existing one is insensitive. Lastly, we apply our proposed *uniformity* metric in the practical scenarios, namely, imposing it as an auxiliary loss term for various existing self-supervised methods, which consistently improves the downstream performance.

The contributions of this work are summarized as: (i) We theoretically and empirically demonstrate the existing uniformity metric (Wang & Isola, 2020) is insensitive to the *dimensional collapse*, and we propose a new uniformity metric that could capture salient sensitivity to the dimensional collapse; (ii) By designing five desirable properties, we open a new perspective to rethink the ideal uniformity metrics; (iii) Our proposed uniformity metric can be applied as an auxiliary loss term in various self-supervised methods, which consistently improves the performance in downstream tasks.

## 2 BACKGROUND

### 2.1 SELF-SUPERVISED REPRESENTATION LEARNING

Self-supervised representation learning aims to learn representations that are invariant to a series of different augmentations. Towards this end, a common practice is to maximize the similarity of representations obtained from different augmented versions of a sample. Specially, given a set of data samples $\{\mathbf{x}_1, \mathbf{x}_2, ..., \mathbf{x}_n\}$, a symmetric network architecture, also called Siamese network (Hadsell et al., 2006), takes as input two randomly augmented views $\mathbf{x}_i^a$ and $\mathbf{x}_i^b$ from a input sample $\mathbf{x}_i$. Then the two views are processed by an encoder network $f$ consisting of a backbone (e.g., ResNet (He et al., 2016)) and a projection MLP head (Chen et al., 2020), denoted as $g$. To enforce invariance to representations of two views $\mathbf{z}_i^a \triangleq g(f(\mathbf{x}_i^a))$ and $\mathbf{z}_i^b \triangleq g(f(\mathbf{x}_i^b))$, a natural solution is to maximize the cosine similarity between representations of two views, and Mean Square Error (MSE) is a widely used loss function to align their $l_2$-normalized representations on the surface of the unit hypersphere:

$$\mathcal{L}_{align}^{\theta} = \|\frac{\mathbf{z}_i^a}{\|\mathbf{z}_i^a\|} - \frac{\mathbf{z}_i^b}{\|\mathbf{z}_i^b\|}\|_2^2 = 2 - 2 \cdot \frac{\langle \mathbf{z}_i^a, \mathbf{z}_i^b \rangle}{\|\mathbf{z}_i^a\| \cdot \|\mathbf{z}_i^b\|} \tag{1}$$

However, a common issue with this approach easily learns an undesired trivial solution that all representations collapse to a constant, as depicted in Fig. 1.

### 2.2 EXISTING SOLUTIONS TO CONSTANT COLLAPSE

To prevent the Siamese network from the constant collapse, existing well-known solutions can be summarized into three types: contrastive learning, asymmetric model architecture, and redundancy reduction. More details will be explained in this section.

**Contrastive Learning** Contrastive learning is one effective way to avoid constant collapse, and the core idea is to repulse negative pairs while attracting positive pairs. SimCLR (Chen et al., 2020) is

one of the most representative works, which first proposes an in-batch negative trick that employs samples in a batch as negative samples. However, its effectiveness heavily relies on the large batch size. To overcome the limitation, MoCo (He et al., 2020) proposes a memory bank to save more representations as negative samples. Besides instance-wise contrastive learning approaches, some recent works also propose clustering-based contrastive learning by bringing together a clustering objective with contrastive learning (Li et al., 2021; Caron et al., 2020).

**Asymmetric Model Architecture**    Asymmetric model architecture is another approach to prevent constant collapse, the core idea is to break the symmetry of the Siamese network. A possible explanation is that asymmetric architecture could encourage encoding more information (Grill et al., 2020). To keep asymmetry, BYOL (Grill et al., 2020) proposes to use an extra predictor in one branch of the Siamese network, and use momentum update and stop-gradient operator in the other branch. An interesting work DINO (Caron et al., 2021) applies this asymmetry in two encoders, and distills knowledge from the momentum encoder to another branch (Hinton et al., 2015). Chen et al. propose SimSiam (Chen & He, 2021) by removing the momentum update from BYOL, its success demonstrates the momentum update is not the key to preventing collapse. Mirror-SimSiam (Zhang et al., 2022a) further swap the stop-gradient operator to the other branch, its failure refutes the claim in SimSiam (Chen & He, 2021), that the stop-gradient operator is the key component to preventing the model from collapse.

**Redundancy Reduction**    The principle for redundancy reduction to prevent constant collapse is to maximize the information content of the representations. The core is to achieve decorrelation by making the matrix based on representations as close to the identity matrix as possible. Barlow Twins (Zbontar et al., 2021) tries to achieve this end on the cross-correlation matrix, while VI-CReg (Bardes et al., 2022) chooses on the covariance matrix. Instead of applying regularization to the matrix, W-MSE(Ermolov et al., 2021) employs a direct way to make the covariance matrix equal to the identity matrix via feature-wise whitening. Zero-CL (Zhang et al., 2022b) further proposes the hybrid of instance-wise and feature-wise whitening to achieve this end.

## 2.3 COLLAPSE ANALYSIS

While aforementioned solutions could effectively prevent model from constant collapse, they might still suffer from the dimensional collapse in which representations occupy a lower-dimensional subspace instead of the entire available embedding space, as depicted in the Fig. 1. The evidence of dimensional collapse was identified in contrastive learning by singular value spectrum of representations (Jing et al., 2022). However, the singular value spectrum is in the form of pictures, making it hard to conduct statistical comparisons among various approaches in terms of collapse analysis.

To gain a quantifiable analysis of collapse degree, Wang et al. propose a formal definition of *uniformity* metric in (Wang & Isola, 2020), via Radial Basis Function (RBF) kernel (Cohn & Kumar, 2007). More specially, given a set of representation vectors $\{\mathbf{z}_1, \mathbf{z}_2, ..., \mathbf{z}_n\}$ ($\mathbf{z}_i \in \mathbb{R}^m$), the *uniformity* metric is defined as follows:

$$\mathcal{L}_{\mathcal{U}} \triangleq \log \frac{1}{n(n-1)/2} \sum_{i=2}^{n} \sum_{j=1}^{i-1} e^{-t\|\frac{\mathbf{z}_i}{\|\mathbf{z}_i\|} - \frac{\mathbf{z}_j}{\|\mathbf{z}_j\|}\|_2^2}, t > 0, \qquad (2)$$

Where $t$ is a fixed parameter (generally $t = 2$). Despite its usefulness, we theoretically and empirically demonstrate this metric is insensitive to the dimensional collapse in Sec. 4.2 and Sec. 4.3.

## 3 A NEW UNIFORMITY METRIC

In this section, we focus on designing a new uniformity metric that could capture salient sensitivity to the dimensional collapse. In Sec. 3.1, we introduce an interesting finding that *the maximum uniformity could be achieved when learned representations obey zero-mean isotropic Gaussian distribution*. To enforce the uniqueness of the ideal distribution, we adopt its $l_2$-normalized form. Interestingly, we theoretically and empirically demonstrate it is an approximately Gaussian distribution in Sec. 3.2. Based on this principle, we propose to use Wasserstein distance between the distribution of learned representations and its ideal distribution as an *uniformity metric* in Sec. 3.3.

### 3.1 Zero-mean Isotropic Gaussian Distribution, The Maximum Uniformity

As shown in Theorem 1, we provide a theorem that states maximum uniformity could be achieved if learned representations obey zero-mean isotropic Gaussian distribution ($\mathbf{Z} \sim \mathcal{N}(\mathbf{0}, \sigma^2 \mathbf{I})$).

**Theorem 1.** *Let a random variable $\mathbf{Z} \sim \mathcal{N}(\mathbf{0}, \sigma^2 \mathbf{I}_m)$ ($\mathbf{Z} \in \mathbb{R}^m$), its $l_2$-normalized form $\mathbf{Y} = \mathbf{Z}/\|\mathbf{Z}\|_2$ uniformly distribute on the surface of a unit hypersphere $\mathcal{S}^{m-1}$. See App. A for the proof.*

However, obeying zero-mean isotropic Gaussian distribution is a *sufficient but not necessary* for an ideal uniformity of its $l_2$-normalized form. For example, as stated in Corollary 1, a mixture of two independent random variables following zero-mean isotropic Gaussian distribution also achieves the ideal uniformity.

**Corollary 1.** *For a random variable $\mathbf{Z}_1, \mathbf{Z}_2 \in \mathbb{R}^m$ that both follow Gaussian distributions. Namely, $\mathbf{Z}_1 \sim \mathcal{N}(\mathbf{0}, \sigma_1^2 \mathbf{I}_m)$, and $\mathbf{Z}_2 \sim \mathcal{N}(\mathbf{0}, \sigma_2^2 \mathbf{I}_m)$. Let $\mathbf{Z}$ be a mixture distribution [1] derived from $\mathbf{Z}_1$ and $\mathbf{Z}_2$ with any binary selection probabilities. Its $l_2$-normalized form $\mathbf{Y} = \mathbf{Z}/\|\mathbf{Z}\|_2$ also uniformly distribute on the surface of the unit hypersphere $\mathcal{S}^{m-1}$.*

The distribution to achieve ideal uniformity (i.e., the mixture of various zero-mean isotropic Gaussian distributions) is not unique due to the mixture form discussed in Corollary 1; in a sense one could define different mixtures of zero-mean isotropic Gaussian distributions, each of which might have different norms encapsulated in $\sigma_1, \cdots, \sigma_k$. Therefore, we turn to investigate the *l2-**normalized form of these zero-mean isotropic Gaussian distributions** [2], see Sec. 3.2.

### 3.2 On the $l_2$-normalized Gaussian distribution

This section will discuss the characteristics regarding a $l_2$-normalized distribution of a Gaussian distribution mixture, which is found to be close to a Gaussian distribution $\mathcal{N}(\mathbf{0}, \frac{1}{m}\mathbf{I}_m)$, from both a theoretical aspect (in Sec. 3.2.1) and an empirical aspect (in Sec. 3.2.2).

#### 3.2.1 Theoretical connection between $\mathbf{Y}$ and the Gaussian distribution

For simplicity, we denote the $l2$-normalized form of zero-mean isotropic Gaussian distributions as $\mathbf{Y}$, $\mathbf{Y} = \mathbf{Z}/\|\mathbf{Z}\|_2$. Note that $\mathbf{Y}$ obeys uniform distribution on the surface of the unit hypersphere $\mathbf{Y} \sim U(\mathcal{S}^{m-1})$. Interestingly, we found $\mathbf{Y}$ approximates a Gaussian distribution $\mathcal{N}(\mathbf{0}, \frac{1}{m}\mathbf{I}_m)$ when $m$ is large enough. Particularly, each dimension of $\mathbf{Y}$, denoted as $Y_i$, *degrades to a Gaussian distribution $\mathcal{N}(0, \frac{1}{m})$* in terms of the Kullback-Leibler divergence when $m$ is large enough, see Theorem 2.

**Theorem 2.** *For a random variable $Y_i$ in the $i$-th dimension of $\mathbf{Y} = \mathbf{Z}/\|\mathbf{Z}\|_2$, where $\mathbf{Z} \sim \mathcal{N}(\mathbf{0}, \sigma^2 \mathbf{I}_m)$ ($\mathbf{Z} \in \mathbb{R}^m$), then the Kullback-Leibler divergence between $Y_i$ and the variable $\hat{Y}_i \sim \mathcal{N}(0, \frac{1}{m})$ converges to zero as $m \to \infty$ as follows.*

$$\lim_{m \to \infty} \mathcal{D}_{KL}(\hat{Y}_i, Y_i) = 0$$

We firstly seek the probability density function (pdf) of $Y_i$ as shown in App. C. Since the probability density functions of both distributions are known, we could derive the Kullback-Leibler divergence between them. One trick is to expand a logarithm term using Taylor expansion. Finally, we obtain that the divergence has a limit of zero when $m$ approaches infinity (Theorem 2 proved). See App. D for the detailed proof.

---

[1] A mixture distribution is the probability distribution of a random variable that is derived from a collection of other random variables. This could be implemented by first sampling a random variable based on a given probability distribution w.r.t a ratio of each random variable, and then sampling a value based on the selected random variable.

[2] Most recent self-supervised representation learning approaches learn representations with a $l_2$ norm constraint (Zbontar et al., 2021; Wang & Isola, 2020; Chen & He, 2021; Grill et al., 2020; Chen et al., 2020), restricting the output representations to the surface of unit hypersphere, i.e., the $l2$-normalized representation ($\mathbf{Y} \overset{def}{=} \mathbf{Z}/\|\mathbf{Z}\|_2$) should be on the surface of the unit hypersphere $\mathcal{S}^{m-1}$. This suggests that directions of learned representation vectors (instead of the absolute amplitude of elements in the vectors) matter when capturing the semantic information of instances.

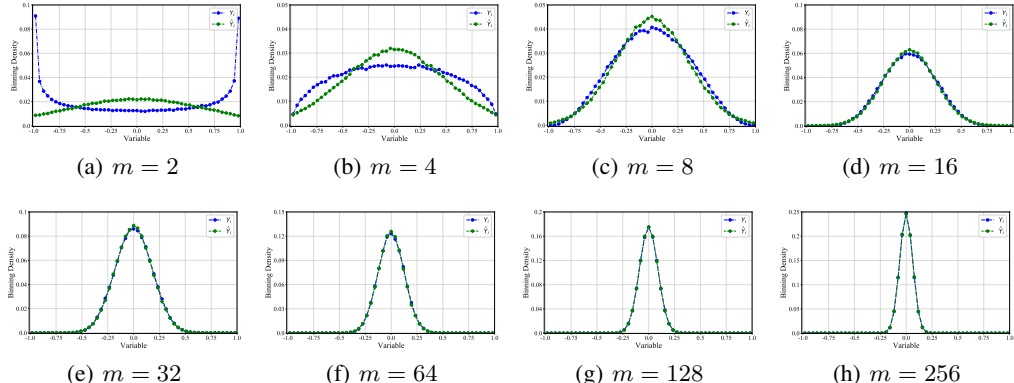

Figure 2: The binning density between $Y_i$ and $\hat{Y}_i$ over various dimensions. See 2d visualization in App. F

Therefore, we make an **assumption** that $l_2$-normalized zero-mean isotropic Gaussian distribution (denoted $\mathbf{Y}$) follows, or at least is close to, an approximated Gaussian distribution $\mathcal{N}(\mathbf{0}, \frac{1}{m}\mathbf{I}_m)$, even $m$ is moderately large. Note that $\mathcal{N}(\mathbf{0}, \frac{1}{m}\mathbf{I}_m)$ enjoys the merits of uniqueness (a proper distribution to design uniformity metric), and might be used as an approximated distribution for $\mathbf{Y}$.

### 3.2.2 EMPIRICAL CONNECTION BETWEEN $\mathbf{Y}$ AND THE GAUSSIAN DISTRIBUTION

The above theoretical conclusion states that the distribution of $\mathbf{Y}$ is infinitely close to a Gaussian distribution when is $m$ infinitely large; while, in practice, we have to adopt finitely large $m$ due to the memory limit. Here we empirically check the closeness between $\mathbf{Y}$ and a Gaussian distribution when using a manageable size of dimension $m$ in practice.

Without losing any generality, we analyze an arbitrary dimension of $\mathbf{Y}$. The distribution of the $i$-th dimension of $\mathbf{Y}$, as denoted as a random variable $Y_i$, is visualized in Fig. 2 by binning 200,000 sampled data points (called 'samples' later) into 51 groups. Fig. 2 shows the distribution difference between $Y_i$ and $\hat{Y}_i$ when selecting $m$ from a manageable internal $[2, 4, 8, 16, 32, 64, 128, 256]$. Note that the difference becomes negligible when $m$ is moderately large (e.g.. $m > 32$) . To quantitatively measure the closeness, Fig. 3 shows the change of the distance (e.g., Wasserstein distance as defined in App. G) between $Y_i$ and $\hat{Y}_i$ with respect to increasing $m$. One could observe that the distance is converged to zero with the large $m$. This also empirically evidences the conclusion in Theorem 2. More details see App. H.

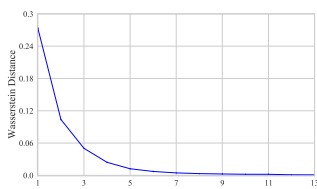

Figure 3: Distance between $Y_i$ and $\hat{Y}_i$

### 3.3 A NEW METRIC FOR UNIFORMITY

In this section, we propose to use the distance between the distribution of learned representations and its ideal Gaussian distribution $\mathcal{N}(\mathbf{0}, \frac{1}{m}\mathbf{I}_m)$ as a uniformity metric. Specially, we collect a set of data vectors from learned representations, i.e., $\{\mathbf{z}_1, \mathbf{z}_2, ..., \mathbf{z}_n\}$, and adopt $l_2$-normalized vectors to calculate the mean and covariance matrix as follows:

$$\boldsymbol{\mu} = \frac{1}{n}\sum_{i=1}^{n}\mathbf{z}_i/\|\mathbf{z}_i\|, \quad \boldsymbol{\Sigma} = \frac{1}{n}\sum_{i=1}^{n}(\mathbf{z}_i/\|\mathbf{z}_i\| - \boldsymbol{\mu})^T(\mathbf{z}_i/\|\mathbf{z}_i\| - \boldsymbol{\mu}), \quad (3)$$

Where $\boldsymbol{\mu} \in \mathbb{R}^m$, $\boldsymbol{\Sigma} \in \mathbb{R}^{m \times m}$, and $m$ is the dimension size of vectors. To facilitate the calculation of distribution distance, we apply a Gaussian hypothesis to learned representations $\mathcal{N}(\boldsymbol{\mu}, \boldsymbol{\Sigma})$. Based on this assumption, we employ Wasserstein distance [3], a well-known distribution distance, to calculate the distance between two distributions, which takes the minimum cost of turning one pile into the other when viewing each distribution as a unit amount of earth/soil, see the definition in App. G.

---

[3]We also discuss using other distribution distances as uniformity metrics, such as Kullback-Leibler Divergence and Bhattacharyya Distance over Gaussian distribution. See more details in App. I.

**Theorem 3.** *Wasserstein Distance (Olkin & Pukelsheim (1982)) Suppose two random variables* $\mathbf{Z}_1 \sim \mathcal{N}(\boldsymbol{\mu}_1, \boldsymbol{\Sigma}_1)$ *and* $\mathbf{Z}_2 \sim \mathcal{N}(\boldsymbol{\mu}_2, \boldsymbol{\Sigma}_2)$ *obey multivariate normal distributions, then* $l_2$*-Wasserstein distance between* $\mathbf{Z}_1$ *and* $\mathbf{Z}_2$ *is:*

$$\mathcal{W}_2(\mathbf{Z}_1, \mathbf{Z}_2) = \sqrt{\|\boldsymbol{\mu}_1 - \boldsymbol{\mu}_2\|_2^2 + Tr(\boldsymbol{\Sigma}_1 + \boldsymbol{\Sigma}_2 - 2(\boldsymbol{\Sigma}_2^{1/2}\boldsymbol{\Sigma}_1\boldsymbol{\Sigma}_2^{1/2})^{1/2})}, \tag{4}$$

Despite its complexity, Wasserstein distance over Gaussian distributions is easy to calculate as illustrated in Theorem 3. We instantiate Equation 4 with the distribution of learned representations and ideal distribution. Then, an *uniformity* metric via Wasserstein distance can be formulated as:

$$\mathcal{W}_2 \triangleq \sqrt{\|\boldsymbol{\mu}\|_2^2 + 1 + Tr(\boldsymbol{\Sigma}) - \frac{2}{\sqrt{m}}Tr(\boldsymbol{\Sigma}^{1/2})}, \tag{5}$$

The smaller $\mathcal{W}_2$, indicates the larger *uniformity* of representations. Besides its usefulness in collapse analysis, our proposed uniformity metric can be also used as an additional loss for various existing self-supervised methods since it is differentiable during the backward pass. One difference is that the mean and covariance matrix in Equation 3 is calculated by batch data during the training phase.

## 4 ON UNIFORMITY METRICS

In this section, we first introduce the desirable properties (called 'Desiderata') of any well-defined uniformity metric in Sec. 4.1. Sec.4.2 and Sec. 4.3 compare the proposed uniformity metric $-\mathcal{W}_2$ with existing uniformity metric $-\mathcal{L}_\mathcal{U}$ theoretically and empirically respectively.

### 4.1 DESIDERATA OF UNIFORMITY

A uniformity metric is a function to map a set of learned representations (typically dense vectors) to a uniformity indicator (typically a real number).

$$\mathcal{U} : \{\mathbb{R}^m\}^n \rightarrow \mathbb{R}, \tag{6}$$

$\mathcal{D} \in \{\mathbb{R}^m\}^n$ is a set of learned vectors ($\mathcal{D} = \{\mathbf{z}_1, \mathbf{z}_2, ..., \mathbf{z}_n\}$), each vector is the feature representation of a instance, $\mathbf{z}_i \in \mathbb{R}^m$. In this section, we formally define five desiderata (i.e., desirable properties) for any uniformity metrics.

Intuitively, *uniformity* is invariant to the permutation of instances, as it cannot affect the distribution.

**Property 1.** *Instance Permutation Constraint (IPC)*

$$\mathcal{U}(\pi(\mathcal{D})) = \mathcal{U}(\mathcal{D}), \tag{7}$$

$\pi$ is an instance permutation operator that changes the order of representations.

The *uniformity* should be invariant when all representations are re-scaled, since modern machine learning tends to use directions of learned representation vectors to capture the semantic information of instances. For example, most recent self-supervised representation learning approaches learn representations with a $l_2$ norm constraint (Zbontar et al., 2021; Wang & Isola, 2020; Grill et al., 2020; Chen et al., 2020), restricting the output representations to the surface of unit hypersphere, i.e., $\mathcal{D}^s = \{\mathbf{s}_1, \mathbf{s}_2, ..., \mathbf{s}_n\}$, and $\mathbf{s}_i = \mathbf{z}_i/\|\mathbf{z}_i\|_2$ is on the surface of the unit hypersphere $\mathcal{S}^{m-1}$.

**Property 2.** *Instance Scaling Constraint (ISC)*

$$\mathcal{U}(\{\lambda_1\mathbf{z}_1, \lambda_2\mathbf{z}_2, ..., \lambda_n\mathbf{z}_n\}) = \mathcal{U}(\mathcal{D}), \quad \forall \lambda_i \in \mathbb{R}^+, \tag{8}$$

The *uniformity* is invariant when instances are cloned, since the cloning operator does not change the original distribution density.

**Property 3.** *Instance Cloning Constraint (ICC)*

$$\mathcal{U}(\mathcal{D} \cup \mathcal{D}) = \mathcal{U}(\mathcal{D}), \tag{9}$$

$\cup$ is the union of two sets that can achieve instance cloning, $\mathcal{D} \cup \mathcal{D} = \{\mathbf{z}_1, \cdots, \mathbf{z}_n, \mathbf{z}_1, \cdots, \mathbf{z}_n\}$.

The *uniformity* decreases when cloning features for each instance, since the feature-level clone will bring some redundancy, leading to dimensional collapse (Zbontar et al., 2021; Bardes et al., 2022).

**Property 4.** *Feature Cloning Constraint (FCC)*

$$\mathcal{U}(\mathcal{D} \oplus \mathcal{D}) \leq \mathcal{U}(\mathcal{D}), \tag{10}$$

$\oplus$ is an feature-level concatenation operator that can achieve feature cloning as $\mathcal{D} \oplus \mathcal{D} = \{\mathbf{z}_1 \oplus \mathbf{z}_1, \mathbf{z}_2 \oplus \mathbf{z}_2, ..., \mathbf{z}_n \oplus \mathbf{z}_n\}$, and where $\mathbf{z}_i \oplus \mathbf{z}_i = [z_{i1}, \cdots, z_{im}, z_{i1}, \cdots, z_{im}]^T \in \mathbb{R}^{2m}$. $\mathcal{U}(\mathcal{D} \oplus \mathcal{D}) = \mathcal{U}(\mathcal{D})$ if and only if $\mathbf{z}_1 = \mathbf{z}_2 = ... = \mathbf{z}_n = \mathbf{0}^m$.

The *uniformity* decreases when adding constant features for each instance, since it introduces uninformative features and results in some collapsed dimensions.

**Property 5.** *Feature Baby Constraint (FBC)*

$$\mathcal{U}(\mathcal{D} \oplus \mathbf{0}^k) \leq \mathcal{U}(\mathcal{D}), \quad k \in \mathbb{N}^+, \tag{11}$$

$\mathcal{D} \oplus \mathbf{0}^k = \{\mathbf{z}_1 \oplus \mathbf{0}^k, \mathbf{z}_2 \oplus \mathbf{0}^k, ..., \mathbf{z}_n \oplus \mathbf{0}^k\}$, and $\mathbf{z}_i \oplus \mathbf{0}^k = [z_{i1}, z_{i2}, ..., z_{im}, 0, 0, ..., 0]^T \in \mathbb{R}^{m+k}$. $\mathcal{U}(\mathcal{D} \oplus \mathbf{0}^k) \leq \mathcal{U}(\mathcal{D})$ if and only if $\mathbf{z}_1 = \mathbf{z}_2 = ... = \mathbf{z}_n = \mathbf{0}^m$.

Note that these five properties are necessary but not sufficient for a well-designed uniformity metric. That is, a well-designed uniformity metric should satisfy these properties; while only satisfying these properties does not sufficiently lead to an ideal uniformity metric.

## 4.2 EXAMINING DESIDERATA OF UNIFORMITY

We employ the desiderata in Sec. 4.1 as criterion to conduct theoretical analysis for two metrics $-\mathcal{L}_\mathcal{U}$ in Equation 2 and $-\mathcal{W}_2$ in Equation 5. The conclusion is stated in the Claim 1 and Claim 2.

**Claim 1.** *Our proposed metric (i.e., $-\mathcal{W}_2$) satisfies all properties including **Property 1, 2, 3, 4**, and **5**. See App. E.1 for the detailed proof.*

**Claim 2.** *the baseline metric (i.e., $-\mathcal{L}_\mathcal{U}$) satisfies **Property 1 and 2**; but it violates **Property 3, 4**, and **5**. See App. E.2 for the detailed proof.*

In terms of *Property IPC* and *Property ISC*, we can directly use their definition to demonstrate both two metrics satisfy the two properties. To further check whether two metric could satisfy other three properties, see App. E for the detailed proof.

Particularly, the proposed metric $-\mathcal{W}_2$ satisfies *FBC Property* while the baseline metric $-\mathcal{L}_\mathcal{U}$ does not. This opens a new angle to explain the advantage of our proposed metric $-\mathcal{W}_2$ from the dimensional collapse perspective. Specially, the larger $k$ would bring the more serious dimensional collapse for $\mathcal{D} \oplus \mathbf{0}^k$ than $\mathcal{D}$. However, $-\mathcal{L}_\mathcal{U}$ fails to identify the more serious dimensional collapse due to $-\mathcal{L}_\mathcal{U}(\mathcal{D} \oplus \mathbf{0}^k) = -\mathcal{L}_\mathcal{U}(\mathcal{D})$. On the contrary, our proposed metric is sensitive to the dimensional collapse as $-\mathcal{W}_2(\mathcal{D} \oplus \mathbf{0}^k) < -\mathcal{W}_2(\mathcal{D})$.

## 4.3 EMPIRICAL ANALYSIS VIA SYNTHETIC DATA

**Correlation between $\mathcal{L}_\mathcal{U}$ and $\mathcal{W}_2$.** We employ synthetic experiments to study uniformity metrics. In detail, we manually sample 50000 data vectors from different distributions, such as standard Gaussian distribution $\mathcal{N}(\mathbf{0}, I)$, uniform Distribution $U(\mathbf{0}, \mathbf{1})$, the mixture of Gaussian, etc. Based on these data vectors, we estimate the uniformity of different distributions by two metrics. As shown in Fig. 4, standard Gaussian distribution achieves the minimum values by both $\mathcal{W}_2$ and $\mathcal{L}_\mathcal{U}$, which indicates that standard Gaussian distribution could achieve larger uniformity than other distributions. This empirical result is consistent with Theorem 1 that standard Gaussian distribution achieves the maximum uniformity.

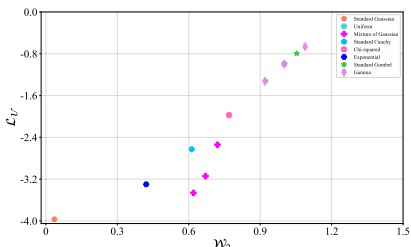

Figure 4: Uniformity analysis on distributions via two metrics.

**On the Dimensional Collapse.** To synthesize data with various specified degrees of dimensional collapse, we concatenate the *zero vectors* (i.e., they are full dimensional collapse) with *sampled data vectors* from the standard Gaussian distribution (i.e., ideal uniformity without collapse). The

percentage of zero-value dimensions of the concatenated vectors is $\eta$ while that of non-zero vectors is $1 - \eta$. As shown in Fig. 5(a) and Fig. 5(b), $\mathcal{W}_2$ is capable of capturing salient sensitivity to collapse level, while $\mathcal{L}_{\mathcal{U}}$ keeps almost no change even in $80\%$ collapse level, indicating $\mathcal{L}_{\mathcal{U}}$ is insensitive to the dimensional collapse.

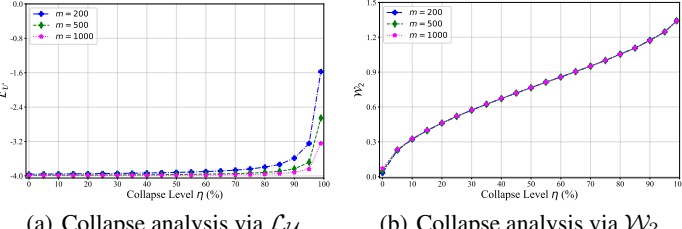

(a) Collapse analysis via $\mathcal{L}_{\mathcal{U}}$      (b) Collapse analysis via $\mathcal{W}_2$

Figure 5: Analysis on dimensional collapse degrees. $\mathcal{W}_2$ is more sensitive to collapse degrees than $\mathcal{L}_{\mathcal{U}}$.

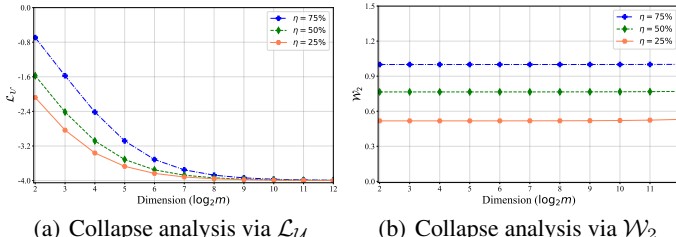

(a) Collapse analysis via $\mathcal{L}_{\mathcal{U}}$      (b) Collapse analysis via $\mathcal{W}_2$

Figure 6: Dimensional collapse w.r.t various dimensions. $\mathcal{L}_{\mathcal{U}}$ fails to identify the dimensional collapse with a large dimension, while $\mathcal{W}_2$ is able to identify the dimensional collapse no matter how great/small $m$ is.

Interestingly, as visualized in Fig. 6, $\mathcal{L}_{\mathcal{U}}$ becomes indistinguishable with different degrees of dimension collapse ($\eta = 25\%, 50\%,$ and $75\%$) when the dimension $m$ becomes large (e.g., $m \geq 2^8$). On the contrary, our proposed $\mathcal{W}_2$ is constant to the dimension number under a specific degree of dimension collapse; $\mathcal{W}_2$ only depends on the degree of dimension collapse and is independent of the dimension number. In summary, our proposed metric $\mathcal{W}_2$ is a more reasonable metric to measure the *uniformity* than the existing one $\mathcal{L}_{\mathcal{U}}$ from an empirical perspective.

## 5 EXPERIMENTS

In this section, we impose the proposed uniformity metric as an auxiliary loss term for various existing self-supervised methods, and conduct experiments on CIFAR-10 and CIFAR-100 datasets to demonstrate its effectiveness. Codes implemented in Pytorch will be released.

**Models** We conduct experiments on a series of self-supervised representation learning models: (i) AlignUniform (Wang & Isola, 2020), whose loss objective consists of an alignment objective and a uniform objective. (ii) three contrastive methods, i.e., SimCLR (Chen et al., 2020), MoCo (He et al., 2020), and NNCLR (Dwibedi et al., 2021). (iii) two asymmetric models, i.e., BYOL (Grill et al., 2020) and SimSiam (Chen & He, 2021). (iv) two methods via redundancy reduction, i.e., BarlowTwins (Zbontar et al., 2021) and Zero-CL (Zhang et al., 2022b). To study the behavior of proposed Wasserstein distance in the self-supervised representation learning, we impose it as an auxiliary loss term to the following models: MoCo v2, BYOL, BarlowTwins, and Zero-CL. To facilitate better use of Wasserstein distance, we also design a linear decay for weighting Wasserstein distance during the training phase, i.e., $\alpha_t = \alpha_{max} - t * (\alpha_{max} - \alpha_{min})/T$, where $t, T, \alpha_{max}$, $\alpha_{min}, \alpha_t$ are current epoch, maximum epochs, maximum weight, minimum weight, and current weight, respectively. More detailed experiments setting see in App. J.

**Metrics** We evaluate the above methods from two perspectives: one is linear evaluation accuracy measured by Top-1 accuracy (Acc@1) and Top-5 accuracy (Acc@5); another is representation capacity. According to (Arora et al., 2019; Wang & Isola, 2020), alignment and uniformity are the two most important properties to evaluate self-supervised representation learning. We use two metrics $\mathcal{L}_{\mathcal{U}}$ and $\mathcal{W}_2$ to measure the uniformity, and a metric $\mathcal{A}$ to measure the alignment between the positive pairs (Wang & Isola, 2020). More details about the alignment metric see in App. K.

Table 1: Main comparison on CIFAR-10 and CIFAR-100 datasets. Proj. and Pred. are the hidden dimension in projector and predictor. ↑ and ↓ mean gains and losses, respectively.

| Methods | Proj. | Pred. | CIFAR-10 | | | | | CIFAR-100 | | | | |
|---|---|---|---|---|---|---|---|---|---|---|---|---|
| | | | Acc@1↑ | Acc@5↑ | $\mathcal{W}_2$↓ | $\mathcal{L_U}$↓ | $\mathcal{A}$↓ | Acc@1↑ | Acc@5↑ | $\mathcal{W}_2$↓ | $\mathcal{L_U}$↓ | $\mathcal{A}$↓ |
| SimCLR | 256 | ✗ | 89.85 | 99.78 | 1.04 | -3.75 | 0.47 | 63.43 | 88.97 | 1.05 | -3.75 | 0.50 |
| NNCLR | 256 | 256 | 87.46 | 99.63 | 1.23 | -3.12 | 0.38 | 54.90 | 83.81 | 1.23 | -3.18 | 0.43 |
| SimSiam | 256 | 256 | 86.71 | 99.67 | 1.19 | -3.33 | 0.39 | 56.10 | 84.34 | 1.21 | -3.29 | 0.42 |
| AlignUniform | 256 | ✗ | 90.37 | 99.76 | 0.94 | -3.82 | 0.51 | 65.08 | 90.15 | 0.95 | -3.82 | 0.53 |
| MoCo v2 | 256 | ✗ | 90.65 | 99.81 | 1.06 | -3.75 | 0.51 | 60.27 | 86.29 | 1.07 | -3.60 | 0.46 |
| MoCo v2 + $\mathcal{W}_2$ | 256 | ✗ | 91.41 ↑0.76 | 99.68 | 0.33 ↑0.73 | -3.84 | 0.63 ↓0.12 | 63.68 ↑3.41 | 88.48 | 0.28 ↑0.79 | -3.86 | 0.66 ↓0.20 |
| BYOL | 256 | 256 | 89.53 | 99.71 | 1.21 | -2.99 | **0.31** | 63.66 | 88.81 | 1.20 | -2.87 | **0.33** |
| BYOL + $\mathcal{W}_2$ | 256 | 256 | 90.31 ↑0.78 | 99.77 | 0.38 ↑0.83 | -3.90 | 0.65 ↓0.34 | 65.16 ↑1.50 | 89.25 | 0.36 ↑0.84 | -3.91 | 0.69 ↓0.36 |
| BarlowTwins | 256 | ✗ | 91.16 | 99.80 | 0.22 | -3.91 | 0.75 | 68.19 | 90.64 | 0.23 | -3.91 | 0.75 |
| BarlowTwins + $\mathcal{W}_2$ | 256 | ✗ | **91.43** ↑0.27 | 99.78 | 0.19 ↑0.03 | -3.92 | 0.76 ↓0.01 | 68.47 ↑0.28 | 90.64 | 0.19 ↑0.04 | -3.91 | 0.79 ↓0.04 |
| Zero-CL | 256 | ✗ | 91.35 | 99.74 | 0.15 | **-3.94** | 0.70 | 68.50 | 90.97 | 0.15 | -3.93 | 0.75 |
| Zero-CL + $\mathcal{W}_2$ | 256 | ✗ | 91.42 ↑0.07 | **99.82** | **0.14** ↑0.01 | **-3.94** | 0.71 ↓0.01 | **68.55** ↑0.05 | **91.02** | **0.14** ↑0.01 | **-3.94** | 0.76 ↓0.01 |

**Main Results** As shown in Tab. 1 We could observe that by imposing $\mathcal{W}_2$ as an additional loss it consistently improves the performance than that without the loss. Interestingly, although it slightly harms alignment, it usually results in improvement in uniformity and finally leads to better accuracy. This demonstrates the effectiveness of $\mathcal{W}_2$ as a uniformity metric. Note imposing an additional loss during training does not affect the training or inference efficiency; therefore, adding $\mathcal{W}_2$ as loss is beneficial without any tangible costs.

**Convergence Analysis** We test the Top-1 accuracy of these models on CIFAR-10 and CIFAR-100 via linear evaluation protocol (as described in App. J) when training them in different epochs. As shown in Fig. 9 in App. L. By imposing $\mathcal{W}_2$ as an additional loss for these models, it converges faster than the raw models, especially for MoCo v2 and BYOL with serious *collapse problem*. Our experiments show that imposing the proposed uniformity metric as an auxiliary penalty loss could largely improve uniformity but damage alignment, see more representation analysis in App. M.

**Dimensional Collapse Analysis** To gain a better understanding of how the additional loss $\mathcal{W}_2$ benefits the alleviation of the dimensional collapse, we visualize singular value spectrum of the representations (Jing et al., 2022). As shown in Fig. 7, the spectrum contains the singular values of the covariance matrix of representations from CIFAR-100 dataset in sorted order and logarithmic scale. Most singular values collapse to zero in BYOL and MoCo v2 models (exclude BarlowTwins), indicating a large number of collapsed dimensions occur in both models. By imposing $\mathcal{W}_2$ as an additional loss for these two models, the number of collapsed dimensions almost decrease to zero, indicating $\mathcal{W}_2$ can effectively address the issue of dimensional collapse.

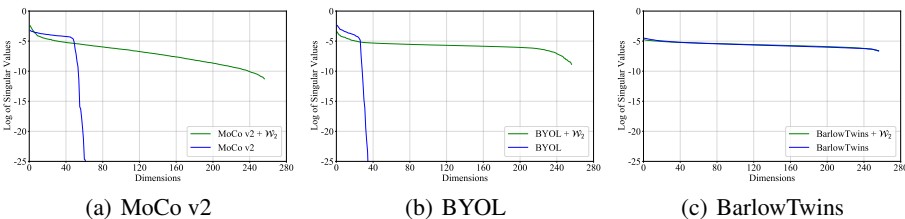

(a) MoCo v2        (b) BYOL        (c) BarlowTwins

Figure 7: Dimensional collapse analysis on CIFAR-100 dataset.

# 6 CONCLUSION

In this paper, we theoretically and empirically demonstrate that the existing uniformity metric is insensitive to the dimensional collapse, and focus on designing a new uniformity metric that could capture salient sensitivity to the dimensional collapse. To this end, we propose to use the Wasserstein distance between the distribution of learned representations and the ideal distribution as the metric of *uniformity*. Furthermore, we formulate five desirable constraints (desiderata) for ideal uniformity metrics, based on which we find that the proposed uniformity metric satisfies all desiderata while the existing one does not. Moreover, we conduct synthetic experiments to further demonstrate that the proposed uniformity metric is capable to deal with the dimensional collapse while the existing one is insensitive. Finally, we apply our proposed metric in the practical scenarios, and impose the proposed *uniformity* metric as an auxiliary loss term for various existing self-supervised methods, which consistently improves the downstream performance. One limitation of our work is that five desirable constraints (desiderata) are not sufficient for ideal uniformity metrics. In future work, we would make further efforts to seek more reasonable properties for uniformity metrics.

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

## A    PROOF OF THE THEOREM 1

*Proof.* According to the property of Gaussian distribution, the distribution of the variable $\mathbf{Z} \sim \mathcal{N}(\mathbf{0}, \sigma^2 \mathbf{I}_m)$ is invariant to arbitrary orthogonal transformation $\mathbf{U}$:

$$\hat{\mathbf{Z}} = \mathbf{U}\mathbf{Z} \sim \mathcal{N}(\mathbf{U}\mathbf{0}, \sigma^2 \mathbf{U}\mathbf{I}_m\mathbf{U}^T) \sim \mathcal{N}(\mathbf{0}, \sigma^2 \mathbf{I}_m) \quad (\mathbf{U}\mathbf{0} = \mathbf{0}, \mathbf{U}\mathbf{I}_m\mathbf{U}^T = \mathbf{U}\mathbf{U}^T = \mathbf{I}_m)$$

Therefore $\hat{\mathbf{Z}}$ is identically distributed with the random variable $\mathbf{Z}$. We denote **i**dentically **d**istributed operation as $\hat{\mathbf{Z}} \overset{id}{\leftrightarrow} \mathbf{Z}$. For the $l_2$-normalized variables:

$$\mathbf{Y} = \mathbf{Z}/\|\mathbf{Z}\|_2, \quad \hat{\mathbf{Y}} = \hat{\mathbf{Z}}/\|\hat{\mathbf{Z}}\|_2, \quad \hat{\mathbf{Y}} \overset{id}{\leftrightarrow} \mathbf{Y}.$$

Since $\|\mathbf{U}\mathbf{Z}\|_2 = \sqrt{(\mathbf{U}\mathbf{Z})^T(\mathbf{U}\mathbf{Z})} = \sqrt{\mathbf{Z}^T\mathbf{U}^T\mathbf{U}\mathbf{Z}} = \sqrt{\mathbf{Z}^T\mathbf{Z}} = \|\mathbf{Z}\|_2$,

$$\hat{\mathbf{Y}} = \frac{\mathbf{U}\mathbf{Z}}{\|\mathbf{U}\mathbf{Z}\|_2} = \frac{\mathbf{U}\mathbf{Z}}{\|\mathbf{Z}\|_2} = \mathbf{U}\mathbf{Y},$$

Therefore, $\mathbf{Y}$ is an identically distributed operation as $\mathbf{U}\mathbf{Y}$, i.e., $\mathbf{Y} \overset{id}{\leftrightarrow} \mathbf{U}\mathbf{Y}$ after an arbitrary orthogonal transformation. To conclude that the random variable $\mathbf{Y}$ uniformly distributes on the surface of the unit hypersphere $\mathcal{S}^{m-1} = \{\mathbf{y} \in \mathbb{R}^m : \|\mathbf{y}\|_2 = 1\}$, here we use **the proof by contradiction**.

Let us assume the opposite of the above conclusion: $\mathbf{Y}$ does not uniformly distribute on the surface of the unit hypersphere $\mathcal{S}^{m-1}$. In other words, the density of each specified-sized area in $\mathbf{Y}$ is not identical for the unit hypersphere $\mathcal{S}^{m-1}$. The random variable $\mathbf{Y}$ has a continuous density $\rho$. Suppose that for $\mathbf{r}_1, \mathbf{r}_2 \in \mathcal{S}^{m-1}$, $\mathbf{r}_1 \neq \mathbf{r}_2$ and $\rho(\mathbf{r}_1) > \rho(\mathbf{r}_2)$, there exists a radius $\epsilon$ for any $l_2$-norm (also holds for other norms) such that on

$$\mathcal{D}_1 = \{\mathbf{r} \in \mathcal{S}^{m-1} : \|\mathbf{r} - \mathbf{r}_1\|_2 < \epsilon\},$$
$$\mathcal{D}_2 = \{\mathbf{r} \in \mathcal{S}^{m-1} : \|\mathbf{r} - \mathbf{r}_2\|_2 < \epsilon\},$$

we still have

$$\forall \mathbf{r} \in \mathcal{D}_1, \forall \mathbf{s} \in \mathcal{D}_2, \rho(\mathbf{r}) > \rho(\mathbf{s}).$$

Therefore, $P(\mathcal{D}_1) > P(\mathcal{D}_2)$. Since $\mathbf{Y} \overset{id}{\leftrightarrow} \mathbf{U}\mathbf{Y}$, $\mathcal{D}_2$ can be obtained from $\mathcal{D}_1$ by a orthogonal transformation [4], which implies that $P(\mathcal{D}_1) = P(\mathcal{D}_2)$.

Contradiction! Hence $\rho(\mathbf{r}_1) = \rho(\mathbf{r}_2)$ for $\forall \mathbf{r}_1, \mathbf{r}_2 \in \mathcal{S}^{m-1}$ and $\mathbf{r}_1 \neq \mathbf{r}_2$. Therefore, $\mathbf{Y} = \mathbf{Z}/\|\mathbf{Z}\|_2$ uniformly distributes on the hypersphere $\mathcal{S}^{m-1}$. $\square$

## B    MEAN AND COVARIANCE MATRIX OF $\mathbf{Y}$

**Theorem 4.** *For a random variable* $\mathbf{Z} \sim \mathcal{N}(\mathbf{0}, \sigma^2 \mathbf{I}_m)$, *and* $\mathbf{Z} \in \mathbb{R}^m$, *for the $l_2$-normalized form* $\mathbf{Y} = \mathbf{Z}/\|\mathbf{Z}\|_2$, *its mean and covariance matrix can be formulated as follows.*

$$\boldsymbol{\mu} = \mathbf{0}, \quad \boldsymbol{\Sigma} = \frac{1}{m}\mathbf{I}_m,$$

*Proof.* $\mathbf{Z} = [z_1, z_2, \cdots, z_m] \sim \mathcal{N}(\mathbf{0}, \sigma^2 \mathbf{I}_m)$, and its probability density function (pdf) can be written as:

$$f_{\mathbf{Z}}(\mathbf{z}) = \frac{1}{(2\pi)^{m/2}|\sigma^2 \mathbf{I}_m|^{1/2}} \exp\{-\frac{1}{2}\mathbf{z}^T(\sigma^2 \mathbf{I}_m)^{-1}\mathbf{z}\} = \frac{1}{(2\pi\sigma^2)^{m/2}} \exp\{-\frac{1}{2}\sum_{i=1}^{m} z_i^2/\sigma^2\},$$

We denote $\mathbf{Y} = [y_1, y_2, \cdots, y_m]$. Then the mean of $i$-th variable $y_i$ can be written as below:

$$\mathbb{E}[y_i] = \int_{z_1}\int_{z_2}\int_{\ldots}\int_{z_m} \frac{z_i}{\sqrt{\sum_i^m z_i^2}} \frac{1}{(2\pi\sigma^2)^{m/2}} \exp\{-\frac{1}{2}\sum_{i=1}^{m} z_i^2/\sigma^2\}dz_1 dz_2 \cdots dz_m,$$

---

[4]Let $\boldsymbol{W}$ be a orthogonal transformation such that $\boldsymbol{W}\boldsymbol{r}_1 = \boldsymbol{r}_2$. $\mathcal{D}_2$ could be obtained by transforming every points from $\mathcal{D}_1$ using orthogonal transformation $\boldsymbol{W}$, namely $\mathcal{D}_2 = \{\boldsymbol{W}\boldsymbol{r} : \boldsymbol{r} \in \mathcal{D}_1\}$,

As $\frac{z_i}{\sqrt{\sum_i^m z_i^2}}$ is an odd function, $\mathbb{E}[y_i] = 0$, and we further conclude $\boldsymbol{\mu} = \mathbb{E}[\mathbf{Y}] = \mathbf{0}$. We also derive the covariance matrix of $\mathbf{Y}$ according to its definition as below:

$$\boldsymbol{\Sigma} = \mathbb{E}[(\mathbf{Y} - \mathbb{E}[\mathbf{Y}])(\mathbf{Y} - \mathbb{E}[\mathbf{Y}])^T] = \begin{pmatrix} \mathbb{E}[y_1^2] & \mathbb{E}[y_1 y_2] & \cdots & \mathbb{E}[y_1 y_m] \\ \mathbb{E}[y_2 y_1] & \mathbb{E}[y_2^2] & \cdots & \mathbb{E}[y_2 y_m] \\ \cdots & \cdots & \cdots & \cdots \\ \mathbb{E}[y_m y_1] & \mathbb{E}[y_m y_2] & \cdots & \mathbb{E}[y_m^2] \end{pmatrix}$$

Then $\mathbb{E}[y_i y_j]$ ($\forall i \neq j$) can be formulated as follows:

$$\mathbb{E}[y_i y_j] = \int_{z_1} \int_{z_2} \int_{...} \int_{z_m} \frac{z_i}{\sqrt{\sum_i^m z_i^2}} \frac{z_j}{\sqrt{\sum_i^m z_i^2}} \frac{1}{(2\pi\sigma^2)^{m/2}} \exp\{-\frac{1}{2}\sum_{i=1}^m z_i^2/\sigma^2\} dz_1 dz_2 \cdots dz_m,$$

As $\frac{z_i}{\sqrt{\sum_i^m z_i^2}} \frac{z_j}{\sqrt{\sum_i^m z_i^2}}$ is an odd function, $\mathbb{E}[y_i y_j] = 0$ ($\forall i \neq j$). In terms of diagonal elements in $\boldsymbol{\Sigma}$, we employ the symmetry to conclude $\mathbb{E}[y_1^2] = \mathbb{E}[y_2^2] = \cdots = \mathbb{E}[y_m^2]$. Based on this principle, we conclude $\mathbb{E}[y_i^2] = \frac{1}{m}$ via below equations:

$$\mathbb{E}[\sum_i^m y_i^2] = m\mathbb{E}[y_i^2], \quad \mathbb{E}[\sum_i^m y_i^2] = \mathbb{E}[\frac{\sum_i^m z_i^2}{\sum_i^m z_i^2}] = 1,$$

Therefore, $\boldsymbol{\Sigma} = \frac{1}{m}\mathbf{I}_m$. $\qquad\qquad\qquad\qquad\qquad\qquad\qquad\qquad\qquad\qquad\qquad\square$

## C   PROBABILITY DENSITY FUNCTION OF $\mathbf{Y}_i$

**Theorem 5.** *For a random variable $\mathbf{Z} \sim \mathcal{N}(\mathbf{0}, \sigma^2 \mathbf{I}_m)$, and $\mathbf{Z} \in \mathbb{R}^m$, for the $l_2$-normalized form $\mathbf{Y} = \mathbf{Z}/\|\mathbf{Z}\|_2$, the probability density function (pdf) of a variable $Y_i$ in the arbitrary dimension is:*

$$f_{Y_i}(y_i) = \frac{\Gamma(m/2)}{\sqrt{\pi}\Gamma((m-1)/2)}(1 - y_i^2)^{(m-3)/2}$$

*Proof.* $\mathbf{Z} = [Z_1, Z_2, \cdots, Z_m] \sim \mathcal{N}(\mathbf{0}, \sigma^2 \mathbf{I}_m)$, then $Z_i \sim \mathcal{N}(0, \sigma^2), \forall i \in [1, m]$. We denote the variable $U = Z_i/\sigma \sim \mathcal{N}(0, 1)$, $V = \sum_{j \neq i}^m (Z_j/\sigma)^2 \sim \mathcal{X}^2(m-1)$, then $U$ and $V$ are independent with each other. For the variable $T = \frac{U}{\sqrt{V/(m-1)}}$, it obeys the Student's t-distribution with $m-1$ degrees of freedom, and its probability density function (pdf) is:

$$f_T(t) = \frac{\Gamma(m/2)}{\sqrt{(m-1)\pi}\Gamma((m-1)/2)}(1 + \frac{t^2}{m-1})^{-m/2}$$

For the variable $Y_i = \frac{Z_i}{\sqrt{\sum_{i=1}^m Z_i^2}} = \frac{Z_i}{\sqrt{Z_i^2 + \sum_{j \neq i}^m Z_j^2}} = \frac{Z_i/\sigma}{\sqrt{(Z_i/\sigma)^2 + \sum_{j \neq i}^m (Z_j/\sigma)^2}} = \frac{U}{\sqrt{U^2 + V}}$, then $T = \frac{U}{\sqrt{V/(m-1)}} = \frac{\sqrt{m-1}Y_i}{\sqrt{1-Y_i^2}}$ and $Y_i = \frac{T}{\sqrt{T^2 + m - 1}}$, the relation between the cumulative distribution function (cdf) of $T$ and that of $Y_i$ can be formulated as follows:

$$F_{Y_i}(y_i) = P(\{Y_i \leq y_i\}) = \begin{cases} P(\{Y_i \leq y_i\}) & y_i \leq 0 \\ P(\{Y_i \leq 0\}) + P(\{0 < Y_i \leq y_i\}) & y_i > 0 \end{cases}$$

$$= \begin{cases} P(\{\frac{T}{\sqrt{T^2+m-1}} \leq y_i\}) & y_i \leq 0 \\ P(\{\frac{T}{\sqrt{T^2+m-1}} \leq 0\}) + P(\{0 < \frac{T}{\sqrt{T^2+m-1}} \leq y_i\}) & y_i > 0 \end{cases}$$

$$= \begin{cases} P(\{\frac{T^2}{T^2+m-1} \geq y_i^2, T \leq 0\}) & y_i \leq 0 \\ P(\{T \leq 0\} + P(\{\frac{T^2}{T^2+m-1} \leq y_i^2, T > 0\}) & y_i > 0 \end{cases}$$

$$= \begin{cases} P(\{T \leq \frac{\sqrt{m-1}y_i}{\sqrt{1-y_i^2}}\}) & y_i \leq 0 \\ P(\{T \leq 0\} + P(\{0 < T \leq \frac{\sqrt{m-1}y_i}{\sqrt{1-y_i^2}}\}) & y_i > 0 \end{cases}$$

$$= P(\{T \leq \frac{\sqrt{m-1}y_i}{\sqrt{1-y_i^2}}\}) = F_T(\frac{\sqrt{m-1}y_i}{\sqrt{1-y_i^2}})$$

Therefore, the pdf of $Y_i$ can be derived as follows:

$$
\begin{aligned}
f_{Y_i}(y_i) &= \frac{d}{dy_i}F_{Y_i}(y_i) = \frac{d}{dy_i}F_T(\frac{\sqrt{m-1}y_i}{\sqrt{1-y_i^2}}) \\
&= f_T(\frac{\sqrt{m-1}y_i}{\sqrt{1-y_i^2}})\frac{d}{dy_i}(\frac{\sqrt{m-1}y_i}{\sqrt{1-y_i^2}}) \\
&= [\frac{\Gamma(m/2)}{\sqrt{(m-1)\pi}\Gamma((m-1)/2)}(1-y_i^2)^{m/2}][\sqrt{m-1}(1-y_i^2)^{-3/2}] \\
&= \frac{\Gamma(m/2)}{\sqrt{\pi}\Gamma((m-1)/2)}(1-y_i^2)^{(m-3)/2}
\end{aligned}
$$

$\square$

## D  PROOF OF THE THEOREM 2

*Proof.* For the variable $\hat{Y}_i \sim \mathcal{N}(0, \frac{1}{m})$, its pdf and $k$-th order raw moment can be formulated as:

$$
f_{\hat{Y}_i}(y) = \sqrt{\frac{m}{2\pi}}\exp\{-\frac{my^2}{2}\}, \quad \mathbb{E}[\hat{Y}_i^k] = \begin{cases} \frac{\prod_{j=1}^{k/2}(2j-1)}{m^j} & k=2j, j=1,2,3... \\ 0 & k=2j-1 \end{cases}
$$

According to the Theorem 5, the pdf of $Y_i$ is:

$$
f_{Y_i}(y_i) = \frac{\Gamma(m/2)}{\sqrt{\pi}\Gamma((m-1)/2)}(1-y_i^2)^{(m-3)/2}
$$

For $0 \le y^2 < 1$, the Taylor expansion of $\log(1-y^2)$ can be written as:

$$
\log(1-y^2) = -\sum_{j=1}^{\infty}\frac{y^{2j}}{j}
$$

Then the Kullback-Leibler divergence between $\hat{Y}_i$ and $Y_i$ can be formulated as:

$$
\begin{aligned}
\mathcal{D}_{KL}(\hat{Y}_i, Y_i) &= \int_{-\infty}^{\infty}f_{\hat{Y}_i}(y)[\log f_{\hat{Y}_i}(y) - \log f_{Y_i}(y_i)]dy \\
&= \int_{-\infty}^{\infty}f_{\hat{Y}_i}(y)[\log\sqrt{\frac{m}{2\pi}} - \frac{my^2}{2} - \log\frac{\Gamma(m/2)}{\sqrt{\pi}\Gamma((m-1)/2)} - \frac{m-3}{2}\log(1-y^2)]dy \\
&= \log\sqrt{\frac{m}{2\pi}} - \log\frac{\Gamma(m/2)}{\sqrt{\pi}\Gamma((m-1)/2)} + \int_{-\infty}^{\infty}f_{\hat{Y}_i}(y)[-\frac{my^2}{2} - \frac{m-3}{2}\log(1-y^2)] \\
&= \log\sqrt{\frac{m}{2}}\frac{\Gamma((m-1)/2)}{\Gamma(m/2)} + \int_{-\infty}^{\infty}f_{\hat{Y}_i}(y)[-\frac{my^2}{2} + \frac{m-3}{2}\sum_{j=1}^{\infty}\frac{y^{2j}}{j}] \\
&= \log\sqrt{\frac{m}{2}}\frac{\Gamma((m-1)/2)}{\Gamma(m/2)} - \frac{m}{2}\mathbb{E}(\hat{Y}_i^2) + \frac{m-3}{2}\sum_{j=1}^{\infty}\mathbb{E}(\hat{Y}_i^{2j})/j \\
&= \log\sqrt{\frac{m}{2}}\frac{\Gamma((m-1)/2)}{\Gamma(m/2)} - \frac{1}{2} + \frac{m-3}{2}[\frac{1}{m} + \frac{3}{2m^2} + \frac{5*3}{3m^3} + o(\frac{1}{m^3})]
\end{aligned}
$$

According to the Stirling formula, we have $\Gamma(x+\alpha) \to \Gamma(x)x^\alpha$ as $x \to \infty$, therefore:

$$
\begin{aligned}
\lim_{m\to\infty}\log\sqrt{\frac{m}{2}}\frac{\Gamma((m-1)/2)}{\Gamma(m/2)} &= \lim_{m\to\infty}\log\sqrt{\frac{m}{2}}\frac{\Gamma((m-1)/2)}{\Gamma((m-1)/2)(\frac{m-1}{2})^{1/2}} \\
&= \lim_{m\to\infty}\log\sqrt{\frac{m}{2}}\sqrt{\frac{2}{m-1}} = 0
\end{aligned}
$$

Then the Kullback-Leibler divergence between $\hat{Y}_i$ and $Y_i$ converges to zero as $m \to \infty$ as follows:

$$\lim_{m\to\infty} \mathcal{D}_{KL}(\hat{Y}_i, Y_i) = \lim_{m\to\infty} \log \sqrt{\frac{m}{2}} \frac{\Gamma((m-1)/2)}{\Gamma(m/2)} - \frac{1}{2} + \frac{m-3}{2}[\frac{1}{m} + \frac{3}{2m^2} + \frac{5*3}{3m^3} + o(\frac{1}{m^3})]$$

$$= 0 + \lim_{m\to\infty} -\frac{1}{2} + \frac{m-3}{2}[\frac{1}{m} + \frac{3}{2m^2} + \frac{5*3}{3m^3} + o(\frac{1}{m^3})] = 0$$

$\square$

# E EXAMINING THE DESIDERATA FOR TWO UNIFORMITY METRICS

## E.1 PROOF FOR $-\mathcal{W}_2$ ON DESIDERATA

The first two properties (**Property** 1 and 2) could be easily proved using the definition. We here to examine the rest three properties one by one for the proposed uniformity metric $-\mathcal{W}_2$.

*Proof.* Firstly, we prove that our proposed metric $-\mathcal{W}_2$ could satisfy the **Property 3**. As $\mathcal{D} \cup \mathcal{D} = \{\mathbf{z}_1, \mathbf{z}_2, ..., \mathbf{z}_n, \mathbf{z}_1, \mathbf{z}_2, ..., \mathbf{z}_n\}$, then its mean vector and covariance matrix can be formulated as follows:

$$\hat{\boldsymbol{\mu}} = \frac{1}{2n} \sum_{i=1}^{n} 2\mathbf{z}_i/\|\mathbf{z}_i\| = \boldsymbol{\mu}, \quad \hat{\boldsymbol{\Sigma}} = \frac{1}{2n} \sum_{i=1}^{n} 2(\mathbf{z}_i/\|\mathbf{z}_i\| - \hat{\boldsymbol{\mu}})^T(\mathbf{z}_i/\|\mathbf{z}_i\| - \hat{\boldsymbol{\mu}}) = \boldsymbol{\Sigma},$$

Then we have:

$$\mathcal{W}_2(\mathcal{D} \cup \mathcal{D}) \triangleq \sqrt{\|\hat{\boldsymbol{\mu}}\|_2^2 + 1 + Tr(\hat{\boldsymbol{\Sigma}}) - \frac{2}{\sqrt{m}}Tr(\hat{\boldsymbol{\Sigma}}^{1/2})} = \mathcal{W}_2(\mathcal{D}).$$

Therefore, $-\mathcal{W}_2(\mathcal{D} \cup \mathcal{D}) = -\mathcal{W}_2(\mathcal{D})$, indicating that our proposed metric $-\mathcal{W}_2$ could satisfy the **Property 3**.

Then, we prove that our proposed metric $-\mathcal{W}_2$ could satisfy the **Property 4**. Given $\mathbf{z}_i = [z_{i1}, z_{i2}, ..., z_{im}]^T$, and $\hat{\mathbf{z}}_i = \mathbf{z}_i \oplus \mathbf{z}_i = [z_{i1}, z_{i2}, ..., z_{im}, z_{i1}, z_{i2}, ..., z_{im}]^T \in \mathbb{R}^{2m}$, for the set: $\mathcal{D} \oplus \mathcal{D}$, its mean vector and covariance matrix can be formulated as follows:

$$\hat{\boldsymbol{\mu}} = \begin{pmatrix} \boldsymbol{\mu}/\sqrt{2} \\ \boldsymbol{\mu}/\sqrt{2} \end{pmatrix}, \quad \hat{\boldsymbol{\Sigma}} = \begin{pmatrix} \boldsymbol{\Sigma}/2 & \boldsymbol{\Sigma}/2 \\ \boldsymbol{\Sigma}/2 & \boldsymbol{\Sigma}/2 \end{pmatrix}$$

As $\hat{\boldsymbol{\Sigma}}^{1/2} = \begin{pmatrix} \boldsymbol{\Sigma}^{1/2}/2 & \boldsymbol{\Sigma}^{1/2}/2 \\ \boldsymbol{\Sigma}^{1/2}/2 & \boldsymbol{\Sigma}^{1/2}/2 \end{pmatrix}$, $Tr(\hat{\boldsymbol{\Sigma}}) = Tr(\boldsymbol{\Sigma})$ and $Tr(\hat{\boldsymbol{\Sigma}}^{1/2}) = Tr(\boldsymbol{\Sigma}^{1/2})$, Then we have,

$$\mathcal{W}_2(\mathcal{D} \oplus \mathcal{D}) \triangleq \sqrt{\|\hat{\boldsymbol{\mu}}\|_2^2 + 1 + Tr(\hat{\boldsymbol{\Sigma}}) - \frac{2}{\sqrt{2m}}Tr(\hat{\boldsymbol{\Sigma}}^{1/2})}$$

$$= \sqrt{\|\boldsymbol{\mu}\|_2^2 + 1 + Tr(\boldsymbol{\Sigma}) - \frac{2}{\sqrt{2m}}Tr(\boldsymbol{\Sigma}^{1/2})},$$

$$> \sqrt{\|\boldsymbol{\mu}\|_2^2 + 1 + Tr(\boldsymbol{\Sigma}) - \frac{2}{\sqrt{m}}Tr(\boldsymbol{\Sigma}^{1/2})} = \mathcal{W}_2(\mathcal{D}),$$

Therefore, $-\mathcal{W}_2(\mathcal{D} \oplus \mathcal{D}) < -\mathcal{W}_2(\mathcal{D})$, indicating that our proposed metric $-\mathcal{W}_2$ could satisfy the **Property 4**.

Finally, we prove that our proposed metric $-\mathcal{W}_2$ could satisfy the **Property 5**. Given $\mathbf{z}_i = [z_{i1}, z_{i2}, ..., z_{im}]^T$, and $\hat{\mathbf{z}}_i = \mathbf{z}_i \oplus \mathbf{0}^k = [z_{i1}, z_{i2}, ..., z_{im}, 0, 0, ..., 0]^T \in \mathbb{R}^{m+k}$, for the set: $\mathcal{D} \oplus \mathbf{0}^k$, its mean vector and covariance matrix can be formulated as follows:

$$\hat{\boldsymbol{\mu}} = \begin{pmatrix} \boldsymbol{\mu} \\ \mathbf{0}^k \end{pmatrix}, \quad \hat{\boldsymbol{\Sigma}} = \begin{pmatrix} \boldsymbol{\Sigma} & \mathbf{0}^{m \times k} \\ \mathbf{0}^{k \times m} & \mathbf{0}^{k \times k} \end{pmatrix}$$

Therefore, $Tr(\hat{\boldsymbol{\Sigma}}) = Tr(\boldsymbol{\Sigma})$, and $Tr(\hat{\boldsymbol{\Sigma}}^{1/2}) = Tr(\boldsymbol{\Sigma}^{1/2})$:

$$\mathcal{W}_2(\mathcal{D} \oplus \mathbf{0}^k) \triangleq \sqrt{\|\hat{\boldsymbol{\mu}}\|_2^2 + 1 + Tr(\hat{\boldsymbol{\Sigma}}) - \frac{2}{\sqrt{m+k}} Tr(\hat{\boldsymbol{\Sigma}}^{1/2})}$$

$$= \sqrt{\|\boldsymbol{\mu}\|_2^2 + 1 + Tr(\boldsymbol{\Sigma}) - \frac{2}{\sqrt{m+k}} Tr(\boldsymbol{\Sigma}^{1/2})}$$

$$> \sqrt{\|\boldsymbol{\mu}\|_2^2 + 1 + Tr(\boldsymbol{\Sigma}) - \frac{2}{\sqrt{m}} Tr(\boldsymbol{\Sigma}^{1/2})} = \mathcal{W}_2(\mathcal{D})$$

Therefore, $-\mathcal{W}_2(\mathcal{D} \oplus \mathbf{0}^k) < -\mathcal{W}_2(\mathcal{D})$, indicating that our proposed metric $-\mathcal{W}_2$ could satisfy the **Property 5**. $\square$

### E.2 PROOF FOR $-\mathcal{L}_{\mathcal{U}}$ ON DESIDERATA

The first two properties (**Property** 1 and 2) could be easily proved using the definition. We here to examine the rest three properties one by one for the existing uniformity metric $-\mathcal{L}_{\mathcal{U}}$.

*Proof.* Firstly, we prove that the baseline metric $-\mathcal{L}_{\mathcal{U}}$ cannot satisfy the **Property 3**. According to the definition of $\mathcal{L}_{\mathcal{U}}$ in Equation 2, we have:

$$\mathcal{L}_{\mathcal{U}}(\mathcal{D} \cup \mathcal{D}) \triangleq \log \frac{1}{2n(2n-1)/2} (4 \sum_{i=2}^{n} \sum_{j=1}^{i-1} e^{-t\|\frac{\mathbf{z}_i}{\|\mathbf{z}_i\|} - \frac{\mathbf{z}_j}{\|\mathbf{z}_j\|}\|_2^2} + \sum_{i=1}^{n} e^{-t\|\frac{\mathbf{z}_i}{\|\mathbf{z}_i\|} - \frac{\mathbf{z}_i}{\|\mathbf{z}_i\|}\|_2^2})$$

$$= \log \frac{1}{2n(2n-1)/2} (4 \sum_{i=2}^{n} \sum_{j=1}^{i-1} e^{-t\|\frac{\mathbf{z}_i}{\|\mathbf{z}_i\|} - \frac{\mathbf{z}_j}{\|\mathbf{z}_j\|}\|_2^2} + n),$$

We set $G = \sum_{i=2}^{n} \sum_{j=1}^{i-1} e^{-t\|\frac{\mathbf{z}_i}{\|\mathbf{z}_i\|} - \frac{\mathbf{z}_j}{\|\mathbf{z}_j\|}\|_2^2}$, and then we have:

$$G = \sum_{i=2}^{n} \sum_{j=1}^{i-1} e^{-t\|\frac{\mathbf{z}_i}{\|\mathbf{z}_i\|} - \frac{\mathbf{z}_j}{\|\mathbf{z}_j\|}\|_2^2} \leq \sum_{i=2}^{n} \sum_{j=1}^{i-1} e^{-t\|\frac{\mathbf{z}_i}{\|\mathbf{z}_i\|} - \frac{\mathbf{z}_i}{\|\mathbf{z}_i\|}\|_2^2} = n(n-1)/2$$

$G = n(n-1)/2$ if and only if $\mathbf{z}_1 = \mathbf{z}_2 = ... = \mathbf{z}_n$.

$$\mathcal{L}_{\mathcal{U}}(\mathcal{D} \cup \mathcal{D}) - \mathcal{L}_{\mathcal{U}}(\mathcal{D}) = \log \frac{4G + n}{2n(2n-1)/2} - \log \frac{G}{n(n-1)/2}$$

$$= \log \frac{(4G+n)n(n-1)/2}{2nG(2n-1)/2} = \log \frac{(4G+n)(n-1)}{4nG - 2G}$$

$$= \log \frac{4nG - 4G + n^2 - n}{4nG - 2G} \geq \log 1 = 0.$$

$\mathcal{L}_{\mathcal{U}}(\mathcal{D} \cup \mathcal{D}) = \mathcal{L}_{\mathcal{U}}(\mathcal{D})$ if and only if $G = n(n-1)/2$, which requires $\mathbf{z}_1 = \mathbf{z}_2 = ... = \mathbf{z}_n$ (an extreme case that all representations collapse to a constant point, as depicted in the Fig. 1). We exclude this extreme case for consideration in the paper, and we have $-\mathcal{L}_{\mathcal{U}}(\mathcal{D} \cup \mathcal{D}) < -\mathcal{L}_{\mathcal{U}}(\mathcal{D})$. Therefore, the baseline metric $-\mathcal{L}_{\mathcal{U}}$ cannot satisfy the **Property 3**.

Then, we prove that the baseline metric $-\mathcal{L}_{\mathcal{U}}$ cannot satisfy the **Property 4**. Given $\mathbf{z}_i = [z_{i1}, z_{i2}, ..., z_{im}]^T$, and $\mathbf{z}_j = [z_{j1}, z_{j2}, ..., z_{jm}]^T$, and we set $\hat{\mathbf{z}}_i = \mathbf{z}_i \oplus \mathbf{z}_i$ and $\hat{\mathbf{z}}_j = \mathbf{z}_j \oplus \mathbf{z}_j$, we have:

$$\mathcal{L}_{\mathcal{U}}(\mathcal{D} \oplus \mathcal{D}) \triangleq \log \frac{1}{n(n-1)/2} \sum_{i=2}^{n} \sum_{j=1}^{i-1} e^{-t\|\frac{\hat{\mathbf{z}}_i}{\|\hat{\mathbf{z}}_i\|} - \frac{\hat{\mathbf{z}}_j}{\|\hat{\mathbf{z}}_j\|}\|_2^2},$$

As $\hat{\mathbf{z}}_i = [z_{i1}, z_{i2}, ..., z_{im}, z_{i1}, z_{i2}, ..., z_{im}]^T$ and $\hat{\mathbf{z}}_j = [z_{j1}, z_{j2}, ..., z_{jm}, z_{j1}, z_{j2}, ..., z_{jm}]^T$, then $\|\hat{\mathbf{z}}_i\| = \sqrt{2}\|\mathbf{z}_i\|$, $\|\hat{\mathbf{z}}_j\| = \sqrt{2}\|\mathbf{z}_j\|$, and $\langle \hat{\mathbf{z}}_i, \hat{\mathbf{z}}_j \rangle = 2\langle \mathbf{z}_i, \mathbf{z}_j \rangle$, we have:

$$\|\frac{\hat{\mathbf{z}}_i}{\|\hat{\mathbf{z}}_i\|} - \frac{\hat{\mathbf{z}}_j}{\|\hat{\mathbf{z}}_j\|}\|_2^2 = 2 - 2\frac{\langle \hat{\mathbf{z}}_i, \hat{\mathbf{z}}_j \rangle}{\|\hat{\mathbf{z}}_i\|\|\hat{\mathbf{z}}_j\|} = 2 - 2\frac{2\langle \mathbf{z}_i, \mathbf{z}_j \rangle}{\sqrt{2}\|\mathbf{z}_i\|\sqrt{2}\|\mathbf{z}_j\|} = \|\frac{\mathbf{z}_i}{\|\mathbf{z}_i\|} - \frac{\mathbf{z}_j}{\|\mathbf{z}_j\|}\|_2^2,$$

Therefore, $-\mathcal{L}_\mathcal{U}(\mathcal{D} \oplus \mathcal{D}) = -\mathcal{L}_\mathcal{U}(\mathcal{D})$, indicating that the baseline metric $-\mathcal{L}_\mathcal{U}$ cannot satisfy the **Property 4**.

Finally, we prove that the baseline metric $-\mathcal{L}_\mathcal{U}$ cannot satisfy the **Property 5**. Given $\mathbf{z}_i = [z_{i1}, z_{i2}, ..., z_{im}]^T$, and $\mathbf{z}_j = [z_{j1}, z_{j2}, ..., z_{jm}]^T$, and we set $\hat{\mathbf{z}}_i = \mathbf{z}_i \oplus \mathbf{0}^k$ and $\hat{\mathbf{z}}_j = \mathbf{z}_j \oplus \mathbf{0}^k$, we have:

$$\mathcal{L}_\mathcal{U}(\mathcal{D} \oplus \mathbf{0}^k) \triangleq \log \frac{1}{n(n-1)/2} \sum_{i=2}^{n} \sum_{j=1}^{i-1} e^{-t\|\frac{\hat{\mathbf{z}}_i}{\|\hat{\mathbf{z}}_i\|} - \frac{\hat{\mathbf{z}}_j}{\|\hat{\mathbf{z}}_j\|}\|_2^2},$$

As $\hat{\mathbf{z}}_i = [z_{i1}, z_{i2}, ..., z_{im}, 0, 0, ..., 0]^T$, and $\hat{\mathbf{z}}_j = [z_{j1}, z_{j2}, ..., z_{jm}, 0, 0, ..., 0]^T$, then $\|\hat{\mathbf{z}}_i\| = \|\mathbf{z}_i\|$, $\|\hat{\mathbf{z}}_j\| = \|\mathbf{z}_j\|$, and $\langle \hat{\mathbf{z}}_i, \hat{\mathbf{z}}_j \rangle = \langle \mathbf{z}_i, \mathbf{z}_j \rangle$, therefore:

$$\|\frac{\hat{\mathbf{z}}_i}{\|\hat{\mathbf{z}}_i\|} - \frac{\hat{\mathbf{z}}_j}{\|\hat{\mathbf{z}}_j\|}\|_2^2 = 2 - 2\frac{\langle \hat{\mathbf{z}}_i, \hat{\mathbf{z}}_j \rangle}{\|\hat{\mathbf{z}}_i\|\|\hat{\mathbf{z}}_j\|} = 2 - 2\frac{\langle \mathbf{z}_i, \mathbf{z}_j \rangle}{\|\mathbf{z}_i\|\|\mathbf{z}_j\|} = \|\frac{\mathbf{z}_i}{\|\mathbf{z}_i\|} - \frac{\mathbf{z}_j}{\|\mathbf{z}_j\|}\|_2^2,$$

Therefore, $-\mathcal{L}_\mathcal{U}(\mathcal{D} \oplus \mathbf{0}^k) = -\mathcal{L}_\mathcal{U}(\mathcal{D})$, indicating that the baseline metric $-\mathcal{L}_\mathcal{U}$ cannot satisfy the **Property 5**. $\square$

## F  A TWO-DIMENSIONAL VISUALIZATION FOR $\mathbf{Y}$ AND $\hat{\mathbf{Y}}$

We also analyze the joint binning density and present 2D joint binning density of two arbitrary individual dimensions, $Y_i$ and $Y_j$ ($i \neq j$) in (a), and $\hat{Y}_i$ and $\hat{Y}_j$ ($i \neq j$) in (b). More details about binning density see in App. H. Even $m$ is relatively small (i.e., 32), it looks that the density of two distributions are close.

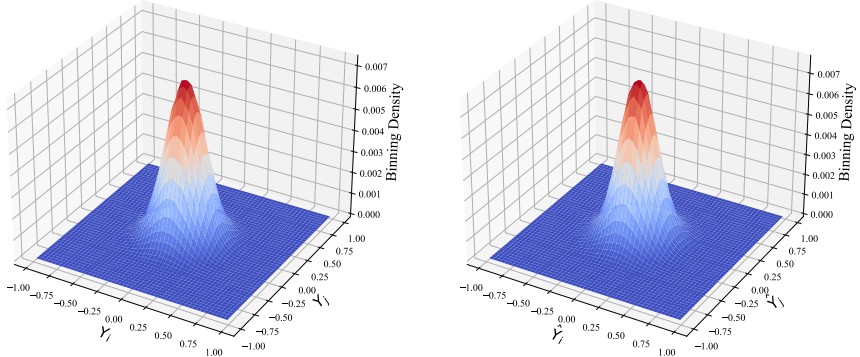

(a) Density for two arbitrary dimensions of $\mathbf{Y}$  (b) Density for two arbitrary dimensions of $\hat{\mathbf{Y}}$

Figure 8: Visualization of two arbitrary dimensions for $\mathbf{Y}$ and $\hat{\mathbf{Y}}$ when $m = 32$. See the binning density in one-dimensional visualization over various dimensions in Fig. 2

## G  THE DEFINITION OF WASSERSTEIN DISTANCE

**Definition 1.** **Wasserstein Distance** or *Earth-Mover* **Distance** with $p$ norm is defined as below:

$$W_p(\mathbb{P}_r, \mathbb{P}_g) = \left(\inf_{\gamma \in \Pi(\mathbb{P}_r, \mathbb{P}_g)} \mathbb{E}_{(x,y) \sim \gamma}[\|x - y\|^p]\right)^{1/p}, \tag{12}$$

where $\Pi(\mathbb{P}_r, \mathbb{P}_g)$ denotes the set of all joint distributions $\gamma(x, y)$ whose marginals are respectively $\mathbb{P}_r$ and $\mathbb{P}_g$. Intuitively, when viewing each distribution as a unit amount of earth/soil, Wasserstein Distance or *Earth-Mover* Distance takes the minimum cost of transporting "mass" from $x$ to $y$ in order to transform the distribution $\mathbb{P}_r$ into the distribution $\mathbb{P}_g$.

## H   DETAILS ON BINNING DENSITY

**Details for 1D Visualization**   The density of $Y_i$ and $\hat{Y}_i$ visualized in Fig. 2 is estimated by binning 200000 data samples into 51 groups. We observe that the density of $Y_i$ would be more overlapped with that of $\hat{Y}_i$. To further verify our observation, we instantiate $\mathbb{P}_r$ and $\mathbb{P}_g$ in Equation 12 with the binning density of $Y_i$ and $\hat{Y}_i$, and employ $W_1(\mathbb{P}_r, \mathbb{P}_g)$ as the distribution distance between $Y_i$ and $\hat{Y}_i$. We calculate $W_1(\mathbb{P}_r, \mathbb{P}_g)$ ten times and average them as visualized in Fig. 3.

**Details for 2D Visualization**   The joint density of $(Y_i, Y_j)$ and $(\hat{Y}_i, \hat{Y}_j)$ $(i \neq j)$, visualized in Fig. 8 is estimated by 2000000 data samples into $51 \times 51$ groups in two-axis ($m = 32$).

## I   OTHER DISTRIBUTION DISTANCES OVER GAUSSIAN DISTRIBUTION

In this section, besides Wasserstein distance over Gaussian distribution, as shown in Theorem 3, we also discuss using other distribution distances as *uniformity* metrics, and make comparisons with Wasserstein distance. As provided Kullback-Leibler Divergence and Bhattacharyya Distance over Gaussian distribution in Theorem 6 and in Theorem 7, both calculations require the covariance matrix is a full rank matrix, making them hard to be used to conduct dimensional collapse analysis. On the contrary, our proposed *uniformity* metric via Wasserstein distance is free from such requirement on the covariance matrix, making it easier to be widely used in practical scenarios.

**Theorem 6.** *Kullback-Leibler Divergence (Lindley & Kullback (1959)) Suppose two random variables $\mathbf{Z}_1 \sim \mathcal{N}(\boldsymbol{\mu}_1, \boldsymbol{\Sigma}_1)$ and $\mathbf{Z}_2 \sim \mathcal{N}(\boldsymbol{\mu}_2, \boldsymbol{\Sigma}_2)$ obey multivariate normal distributions, then Kullback-Leibler divergence between $\mathbf{Z}1$ and $\mathbf{Z}_2$ is:*

$$\mathcal{D}_{KL}(\mathbf{Z}_1, \mathbf{Z}_2) = \frac{1}{2}((\boldsymbol{\mu}_1 - \boldsymbol{\mu}_2)^T \boldsymbol{\Sigma}_2^{-1}(\boldsymbol{\mu}_1 - \boldsymbol{\mu}_2) + Tr(\boldsymbol{\Sigma}_2^{-1}\boldsymbol{\Sigma}_1 - \mathbf{I}) + \ln \frac{\det \boldsymbol{\Sigma}_2}{\det \boldsymbol{\Sigma}_1}),$$

**Theorem 7.** *Bhattacharyya Distance (Bhattacharyya (1943)) Suppose two random variables $\mathbf{Z}_1 \sim \mathcal{N}(\boldsymbol{\mu}_1, \boldsymbol{\Sigma}_1)$ and $\mathbf{Z}_2 \sim \mathcal{N}(\boldsymbol{\mu}_2, \boldsymbol{\Sigma}_2)$ obey multivariate normal distributions, $\boldsymbol{\Sigma} = \frac{1}{2}(\boldsymbol{\Sigma}_1 + \boldsymbol{\Sigma}_2)$, then bhattacharyya distance between $\mathbf{Z}1$ and $\mathbf{Z}_2$ is:*

$$\mathcal{D}_B(\mathbf{Z}_1, \mathbf{Z}_2) = \frac{1}{8}(\boldsymbol{\mu}_1 - \boldsymbol{\mu}_2)^T \boldsymbol{\Sigma}^{-1}(\boldsymbol{\mu}_1 - \boldsymbol{\mu}_2) + \frac{1}{2}\ln \frac{\det \boldsymbol{\Sigma}}{\sqrt{\det \boldsymbol{\Sigma}_1 \det \boldsymbol{\Sigma}_2}},$$

## J   EXPERIMENTS SETTING IN THE EXPERIMENTS

**Setting**   To make a fair comparison, we conduct all experiments in Sec. 5 on a single 1080 GPU. Also, we adopt the same network architecture for all models, i.e., ResNet-18 (He et al., 2016) as the encoder, a three-layer MLP as the projector, and a three-layer MLP as the projector, respectively. Besides, We use LARS optimizer (You et al., 2017) with a base learning rate $0.2$, along with a cosine decay learning rate schedule (Loshchilov & Hutter, 2017) for all models. We evaluate all models under a linear evaluation protocol. In specific, models are pre-trained for 500 epochs and evaluated by adding a linear classifier and training the classifier for 100 epochs while keeping the learned representations unchanged. We also deploy the same augmentation strategy for all models, which is the composition of a series of data augmentation operations, such as color distortion, rotation, and cutout. Following (da Costa et al., 2022), we set temperature $t = 0.2$ for all contrastive methods. As for MoCo (He et al., 2020) and NNCLR (Dwibedi et al., 2021) that require an extra queue to save negative samples, we set the queue size to $2^{12}$. For the linear decay for weighting Wasserstein distance, detailed parameter settings are shown in Table 2.

Table 2: Parameter setting for various models in experiments.

| Models | MoCo v2 | BYOL | BarlowTwins | Zero-CL |
|---|---|---|---|---|
| $\alpha_{max}$ | 1.0 | 0.2 | 30.0 | 30.0 |
| $\alpha_{min}$ | 1.0 | 0.2 | 0 | 30.0 |

## K  ALIGNMENT METRIC FOR SELF-SUPERVISED REPRESENTATION LEARNING

As one of the important indicators to evaluate representation capacity, the alignment metric measures the distance among semantically similar samples in the representation space, and smaller alignment generally brings better representation capacity. Wang et al (Wang & Isola, 2020) propose a simpler approach by calculating the average distance between the positive pairs as alignment, and it can be formulated as follows:

$$\mathcal{A} \triangleq \mathbb{E}_{(\mathbf{z}_i^a, \mathbf{z}_i^b) \sim p_{\mathbf{z}}^{pos}} [\| \frac{\mathbf{z}_i^a}{\|\mathbf{z}_i^a\|} - \frac{\mathbf{z}_i^b}{\|\mathbf{z}_i^b\|} \|_2^\beta] \tag{13}$$

Where $(\mathbf{z}_i^a, \mathbf{z}_i^b)$ is a positive pair as discussed in Sec 2.1. We set $\beta = 2$ in the experiments.

## L  CONVERGENCE ANALYSIS ON TOP-1 ACCURACY

Here we show the change of Top-1 accuracy through all the training epochs in Fig 9. During training, we take the model checkpoint after finishing each epoch to train linear classifier, and then evaluate the Top-1 accuracy on the unseen images of the test set (in either CIFAR-10 or CIFAR-100 ). In both CIFAR-10 and CIFAR-100, we could obverse that imposing the proposed uniformity metric as an auxiliary penalty loss could largely improve the Top-1 accuracy, especially in the early stage.

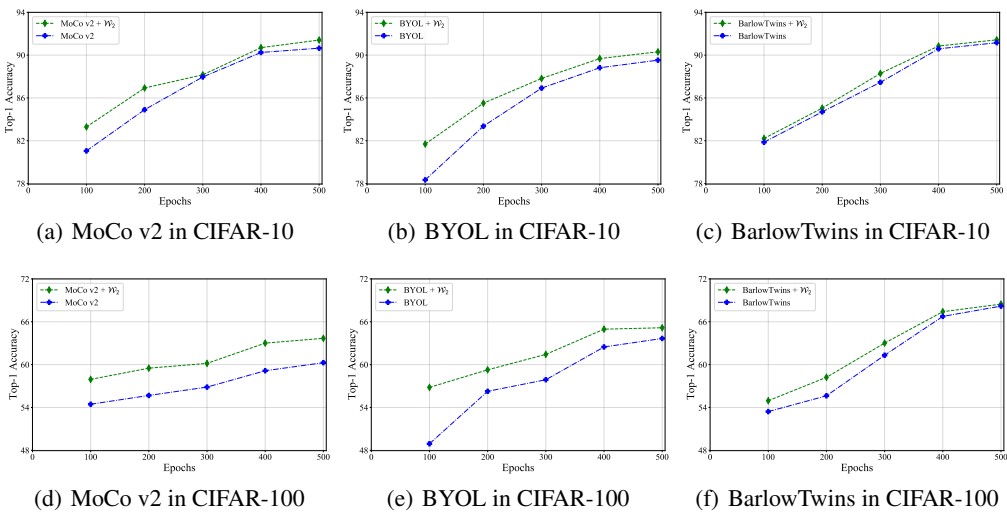

(a) MoCo v2 in CIFAR-10    (b) BYOL in CIFAR-10    (c) BarlowTwins in CIFAR-10

(d) MoCo v2 in CIFAR-100    (e) BYOL in CIFAR-100    (f) BarlowTwins in CIFAR-100

Figure 9: Convergence analysis on Top-1 accuracy during training.

## M  ANALYSIS ON UNIFORMITY AND ALIGNMENT

Here we show the change of uniformity and alignment through all the training epochs in Fig. 10 and Fig. 11 respectively. During training, we take the model checkpoint after finishing each epoch to evaluate the uniformity (i.e., using the proposed metric $\mathcal{W}_2$ ) and alignment (Wang & Isola, 2020) on the unseen images of the test set (in either CIFAR-10 or CIFAR-100 ). In both CIFAR-10 and CIFAR-100, we could obverse that imposing the proposed uniformity metric as an auxiliary penalty loss could largely improve its uniformity. Consequently, it also lightly damage the alignment (*the smaller, the better-aligned*) since a better uniformity usually leads to worse alignment by definition.

## N  THE EXPLANATION FOR PROPERTY 5

Here we explain why the Property 5 is an inequality instead of an equality by case study. Suppose a set of data vectors ($\mathcal{D}$) defined in Sec. 4.1 is with the maximum uniformity. When more dimensions with zero-value are inserted to $\mathcal{D}$, the set of new data vectors ($\mathcal{D} \oplus \mathbf{0}^k$) cannot achieve maximum uniformity any more, as they only occupy a small space on the surface of unit hypersphere. Therefore, the uniformity would decrease significantly with large $k$.

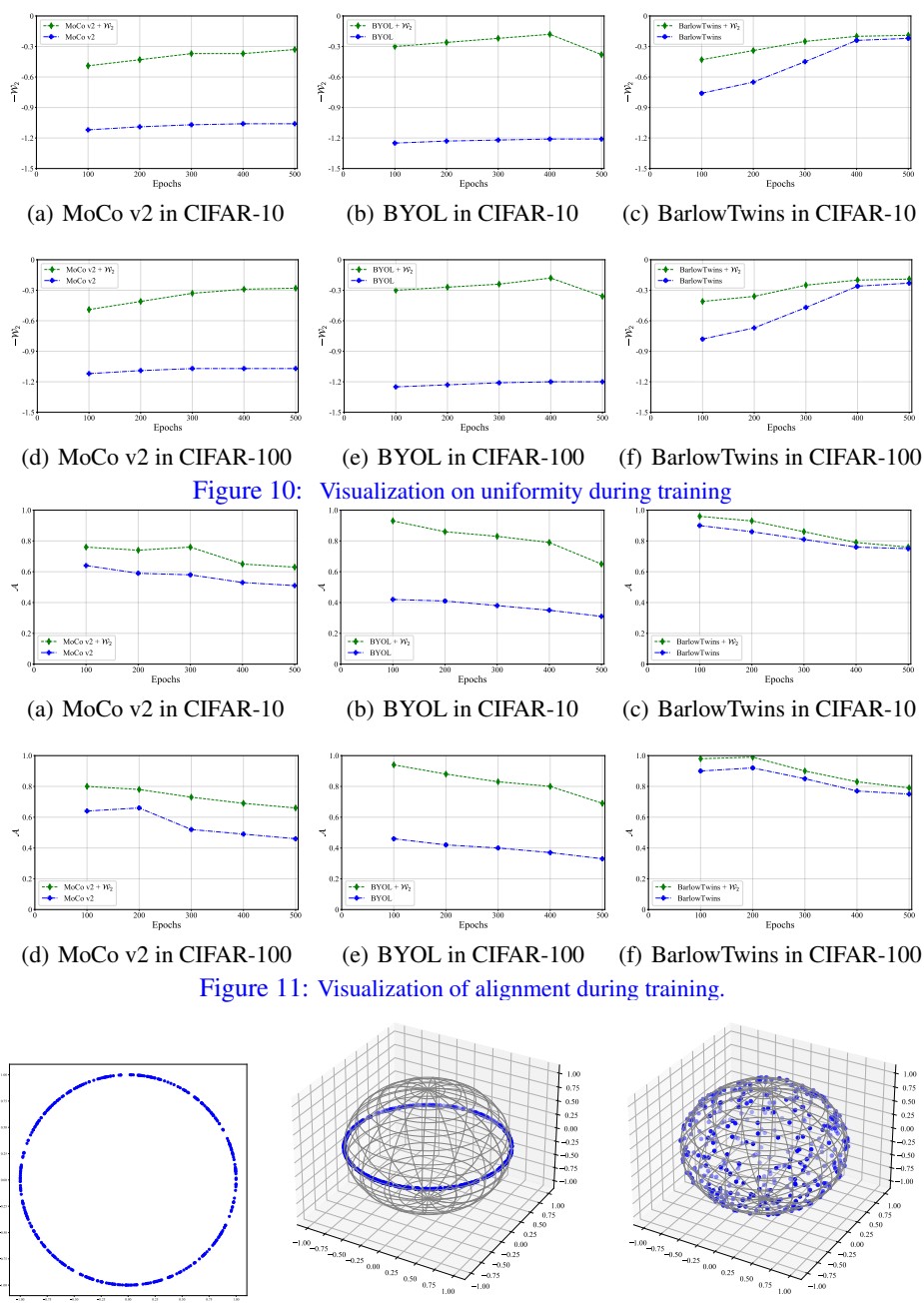

(a) MoCo v2 in CIFAR-10    (b) BYOL in CIFAR-10    (c) BarlowTwins in CIFAR-10

(d) MoCo v2 in CIFAR-100    (e) BYOL in CIFAR-100    (f) BarlowTwins in CIFAR-100

Figure 10: Visualization on uniformity during training

(a) MoCo v2 in CIFAR-10    (b) BYOL in CIFAR-10    (c) BarlowTwins in CIFAR-10

(d) MoCo v2 in CIFAR-100    (e) BYOL in CIFAR-100    (f) BarlowTwins in CIFAR-100

Figure 11: Visualization of alignment during training.

(a) Two-dimensional visual- (b) Three-dimensional visualiza- (c) Three-dimensional visualiza-
ization with no collapsed di- tion with one collapsed dimension tion with no collapsed dimension
mension

Figure 12: Case study for Property 5 and blue point are data vectors.

To further illustrate the inequality, we visualize sampled data vectors. In Fig. 12(a), we visualize 400 data vectors ($\mathcal{D}_1$) sampled from $\mathcal{N}(\mathbf{0}, \mathbf{I}_2)$, and they almost uniformly distribute on the $\mathcal{S}^1$. We insert one dimension with zero-value to $\mathcal{D}_1$, and denote it as $\mathcal{D}_1 \oplus \mathbf{0}^1$, as shown in Fig. 12(b). In comparison with $\mathcal{D}_2$ where 400 data vectors are sampled from $\mathcal{N}(\mathbf{0}, \mathbf{I}_3)$, as visualized in Fig. 12(c), $\mathcal{D}_1 \oplus \mathbf{0}^1$ only occupy a ring on the $\mathcal{S}^2$, while $\mathcal{D}_2$ almost uniformly distribute on the $\mathcal{S}^2$. Therefore, $\mathcal{U}(\mathcal{D}_2) > \mathcal{U}(\mathcal{D}_1 \oplus \mathbf{0}^1)$. Note that no matter how great/small $m$, the baseline uniformity metric (Wang & Isola, 2020) and our proposed uniformity metric have equal maximum uniformity, i.e., $\mathcal{W}_2 = 0$ and $\mathcal{L}_{\mathcal{U}} = -4.0$. Therefore, the maximum uniformity over various dimensions $m$ should be equal, or at least close to, then we have $\mathcal{U}(\mathcal{D}_1) \approx \mathcal{U}(\mathcal{D}_2) > \mathcal{U}(\mathcal{D}_1 \oplus \mathbf{0}^1)$. The Property 5 should be an inequality, and can be used to identify the capacity on capturing sensitivity to the dimensional collapse.

