# OpenReview forum: "Rethinking Uniformity in Self-Supervised Representation Learning"
_ICLR.cc/2023/Conference — Submitted to ICLR 2023_

### Official Review · Reviewer_ZorS · 2022-10-23

**Confidence:** 3
**Correctness:** 4
**Technical Novelty And Significance:** 2
**Empirical Novelty And Significance:** 3
**Recommendation:** 5

**Clarity, Quality, Novelty And Reproducibility:**

The paper is clearly written. To avoid dimensionality collapse, the Wasserstein-based measure is used. This is a practical approach. However, The proposed method is based on assumptions and approximations, and the theoretical analysis is insufficient.

**Strength And Weaknesses:**

Strength
- The readability of the Introduction and related works is excellent. The idea of the proposed method is easy to follow.

Weaknesses
- In section 3, some theorems are presented. Since these theorems are elemental results, theorems should be moved to the appendix. Instead, the authors could discuss more important issues, such as the theoretical properties of the proposed Wasserstein-based uniformity measure.

- The Wasserstein distance from the empirical distribution to the Gaussian distribution with mean zero and variance I/m is employed to measure the uniformity on the unit sphere. To derive the uniformity measure, some assumptions and approximations are introduced. Therefore, the theoretical properties of the proposed measure are not very clear. Even when the data is not uniformly distributed, the Wasserstein-based measure, i.e., the negative Wassersteindeitance, can take a large value.

- The authors should concentrate on thorough theoretical analysis to guarantee the effectiveness of the proposed method.
  - The measure -W_2 takes the first and second-order moment into account. How the higher order moment affects the W_2-based measure (4)?
  - Is it possible to analyze the generalization error of downstream tasks when the proposed measure is used for feature learning?




**Summary Of The Paper:**

This paper studies the dimensionality collapse in self-supervised learning (SSL). Though the uniformity of the representation is regarded as an important feature for downstream tasks, the existing loss function in SimCLR is insensitive to dimensional collapse. With a theoretical guarantee, the authors propose a Wasserstein-based uniformity measure to avoid dimensionality collapse. Numerical experiments show that the proposed method efficiently works to extract useful features for downstream tasks.

**Summary Of The Review:**

The proposed method is based on some assumptions and approximations, and the theoretical analysis is insufficient.

---

> ### Author Response · Authors · 2022-11-14
> **To Reviewer ZorS (Part 1)**
>
> **R 4.1 In section 3, some theorems are presented. Since these theorems are elemental results, theorems should be moved to the appendix. Instead, the authors could discuss more important issues, such as the theoretical properties of the proposed Wasserstein-based uniformity measure.**
>
> As suggested by the respected reviewer, we have substantially revised Sec.3. Especially, we largely shorten the statements regarding Theorem 1 (include statements and proofs) in the main text, see more concise Sec.3.1.  In Sec.3.2 we have moved some obvious statements to the appendix.
>
> By doing so, we could leverage more space to provide more theoretical results and detailed empirical results on the closeness between $Y$ and a Gaussian distribution, which we believe is essential to serve as the foundation of our method.
>
> In the revised version, our discussion on theoretical properties is clearer, and can be summarized into two parts: (i) we provide theoretical evidence to prove that $Y_i$ approximates a Gaussian distribution $\mathcal{N}(0, \frac{1}{m})$ when  $m$ is large enough, see in Theorem 2 in Sec.3.2.1;  (ii)  we also theoretically demonstrate the baseline uniformity metric in [1] is insensitive to the dimensional collapse, see in Claim 2 in Sec.4.2, and demonstrate our proposed uniformity metric could capture salient sensitivity to dimensional collapse, see in Claim 1 in Sec.4.2.
>
> As you may notice the theoretical property (i), we first seek the probability density function (pdf) of $Y_i$ as shown in App.C. Since the probability density functions of both distributions are known, we could derive the  Kullback-Leibler divergence between them. One trick is to expand a logarithm term using Taylor expansion. Finally,  we obtain that the divergence has a limit of zero when $m$ approaches infinity. See App.D for detailed proof.
>
> [1] Tongzhou Wang and Phillip Isola. Understanding contrastive representation learning through alignment and uniformity on the hypersphere. In ICML, 2020
>
> **R 4.2 The Wasserstein distance from the empirical distribution to the Gaussian distribution with mean zero and variance I/m is employed to measure the uniformity on the unit sphere. To derive the uniformity measure, some assumptions and approximations are introduced. Therefore, the theoretical properties of the proposed measure are not very clear. Even when the data is not uniformly distributed, the Wasserstein-based measure, i.e., the negative Wassersteindeitance, can take a large value.**
>
> Thanks for your comment. As discussed above, our theoretical analysis can be summarized into two parts. Besides the theoretical efforts, we add more empirical analysis to support the Gaussian assumption. (i) See the 1D visualization in Fig.2, we analyze the density of $l_2$ normalized Gaussian distribution, and Gaussian distribution over various dimensions, the difference between the two distributions becomes negligible when $m>32$. (ii) we use Wasserstein distance to further quantitatively measure the closeness between two distributions, as visualized in Fig.3, the distance is converged to zero with the large $m$. (iii) We also provide 2D visualization in Fig.8, which also shows the density of the two distributions are close.
>
>
> As you mention that the negative Wasserstein distance might take a large value when the data is not uniformly distributed, we believe this case would not occur. In Sec.4.3, we design representations with different dimensional collapse degrees, by concatenating the zero vectors (i.e., they are full dimensional collapse)  with  sampled data vectors from the standard Gaussian distribution (i.e., ideal uniformity without collapse). See in Fig.5(b), the negative Wasserstein distance monotonically decreases with increasing collapse levels. Even the data is not uniformly distributed, negative Wasserstein distance would take a proper value according to the collapse level no matter how great/small the dimension ($m$) is, as shown in Fig.6(b).

---

> ### Author Response · Authors · 2022-11-14
> **To Reviewer ZorS (Part 2)**
>
> **R 4.3 Proposed measure takes the first and second-order moment into account. How the higher order moment affects the proposed measure**
>
> Based on Gaussian distribution, there is no need to calculate Wasserstein distance by using higher order moment (the proof is demonstrated in [2]). Based on non-Gaussian distribution, the calculation of Wasserstein distance remains unclear. Anyway, we leave the use of high-order moments as future work.
>
> [2] A class of Wasserstein metrics for probability distributions.
>
> **R 4.4 Is it possible to analyze the generalization error of downstream tasks**
>
> We guess we have provided the analysis of the generalization error of downstream tasks. We follow the standard linear evaluation of self-supervised learning: (i) We first pre-train the model, and our proposed uniformity metric is used as an additional loss to improve representation learning without label supervision (The pre-train is on the training dataset). (ii) After pre-training, we freeze the network parameter and only train a newly introduced linear layer with label information (The training is on the training dataset). The downstream performance including the generalization error (Top-1 accuracy and Top-5 on the test dataset) is shown on Tab.1. We also conduct convergence analysis on the accuracy, alignment, and uniformity terms, see in Sec.5
>
> if we misunderstood your comment, we kindly ask if you could further describe the question, and we will carefully address your concern in the next round.
>
> **R 4.5 Elemental theorems in Sec.3 should be move to the Appendix**
>
> We largely shorten the content regarding Theorem 1 (include statements and proofs) in the main text, see more concise Sec 3.1. By doing so, we could leverage more space to highlight the core contributions of this work.
>
> In Sec. 3.2 we have moved some obvious statements and theorems to the appendix. Moreover, we provide more theoretical results and detailed empirical results on the closeness between $Y$ and a Gaussian distribution, which we believe this part is essential to serve as the foundation of our method.
>
>
> We have made the above statements clearer in the revised version, thanks for your constructive comments.

---

### Official Review · Reviewer_iVeQ · 2022-10-24

**Confidence:** 4
**Correctness:** 3
**Technical Novelty And Significance:** 3
**Empirical Novelty And Significance:** 2
**Recommendation:** 6

**Clarity, Quality, Novelty And Reproducibility:**

Clarity: The paper was organized and written well with some minor problems.
Suggested Corrections:

Remove the extra "on the"
… on the on the surface of the unit …

We instantiate Equation 11 with the distribution … There is no equation 11.

The last inequality in Property 5 is supposed to be an equality

Some of the lemma (Lemma 2) and theorems were textbook level concepts. Proofs may not be needed even in the supplementary.

Quality:

The quality of the paper was OK. Experimental results are somewhat strong. However, the experimental results do not confirm whether the improvement is due to network learning better representations or due to improved convergence and shift of the learning curve.

Other aspects of uniformity that would not be satisfied by the proposed metric were not discussed. It was not clear whether these five properties were sufficient to confirm uniformity.


Novelty:

DINO, one of the key self-supervised learning work of recent years was not discussed in the paper. This could have been categorized under "Asymmetric Model Architecture" in related work.

Caron, M., Touvron, H., Misra, I., Jégou, H., Mairal, J., Bojanowski, P., & Joulin, A. (2021). Emerging properties in self-supervised vision transformers. In Proceedings of the IEEE/CVF International Conference on Computer Vision (pp. 9650-9660).

Reproducibility:

Code is not provided. Given the scope of experiments spanning various techniques it is hard to figure if the results would be reproducible.

**Strength And Weaknesses:**

Strengths:
	- The emphasis on dimensional collapse and how it was overlooked by earlier work was well thought
	- The metric is very simple to implement because it only involves evaluating the Wasserstein distance of two Gaussian distributions
	- Extensive experiments have been performed reimplementing several of the recently introduced techniques with the modified loss function.

Weaknesses:
	- The paper assumes that l2-normalized zero-mean isotropic Gaussian distribution follows a Gaussian-like distribution. The illustration in 1D is helpful, but it would have been more interesting to see the deviation trend as the dimensionality increases.
	- Five properties were introduced. It was not discussed whether these five properties were sufficient to prove ideal uniformity.

---------------
Authors' responses somewhat address my critiques but not at a level to increase my rating.

**Summary Of The Paper:**

This paper proposes the Wasserstein distance between an isotropic zero mean Gaussian distribution and the distribution of the learned representations as a robust uniformity metric for self-supervised learning. Five desirable properties of uniformity are discussed and the proposed metric is compared against another recently introduced uniformity metric (Wang & Isola, 2020) in terms of how well the two metrics satisfy these properties. The proposed metric is also added to the loss function and several recent self-supervised algorithms have been retrained using this modified loss function. Results suggest that adding this new uniformity metric-based loss increases accuracy and improves convergence of the training.

**Summary Of The Review:**

The proposed uniformity metric can be considered as an important contribution of the work for the self-supervised learning literature. However, the fact that limitations and potential pitfalls were not discussed  raises important questions about whether the reported improvements are indeed due to the proposed metric or some other aspect of learning is being affected by the modified loss functions.

---

> ### Author Response · Authors · 2022-11-14
> **To Reviewer iVeQ**
>
> **R 3.1  The paper assumes that l2-normalized zero-mean isotropic Gaussian distribution follows a Gaussian-like distribution. The illustration in 1D is helpful, but it would have been more interesting to see the deviation trend as the dimensionality increases.**
>
> Thanks for your comment on the Gaussian assumption. In the revised version, we make theoretical and empirical efforts to support this Gaussian assumption. (i) we provide theoretical evidence to prove that $Y_i$ approximates a Gaussian distribution $\mathcal{N}(0, \frac{1}{m})$ when $m$ is large enough, see in Theorem 2 in Sec.3.2.1; (ii) we provide empirical efforts to check how the closeness between $Y$ and a Gaussian distribution, see in Sec.3.2.2.
>
> As for theoretical evidence, we first seek the probability density function (pdf) of $Y_i$ as shown in App.C. Since the probability density functions of both distributions are known, we could derive the  Kullback-Leibler divergence between them. One trick is to expand a logarithm term using Taylor expansion. Finally,  we obtain that the divergence has a limit of zero when $m$ approaches infinity. See App.D for detailed proof.
>
> As for empirical validation, we add more analysis to support the Gaussian assumption. (i) See the 1D visualization in Fig.2, we analyze the density of $l_2$ normalized Gaussian distribution, and Gaussian distribution over various dimensions, the difference between the two distributions becomes negligible when $m>32$.  (ii) We also provide 2D visualization in Fig.8, which also shows the density of the two distributions are close.
>
> As you mention it would be more interesting to see the deviation trend as the dimensionality increases, we use Wasserstein distance to quantitatively measure the closeness between two distributions, as visualized in Fig.3, the distance is converged to zero with the large $m$.
>
> **R 3.2  Five properties were introduced. It was not discussed whether these five properties were sufficient to prove ideal uniformity**
>
> Thanks for your comment on properties. It is a good question, our proposed properties are necessary but not sufficient, and we have made this statement clear in the last paragraph of Sec. 4.1. We would make further efforts to seek more sufficient and complete properties for uniformity metrics in future work.
>
>
> **R 3.3 The last inequality in Property 5 is supposed to be an equality**
>
> Thanks for your comment on Property 5. Property 5 should be an inequality. Suppose a set of data vectors ($\mathcal{D}$) defined in Sec.4.1 is with the maximum uniformity. When more dimensions with zero-value are inserted to $\mathcal{D}$, the set of new data vectors ($\mathcal{D} \oplus \mathbf{0}^{k}$) cannot achieve maximum uniformity anymore, as they only occupy a small space on the surface of the unit hypersphere. Therefore, the uniformity would decrease significantly with large $k$. We also draw figures in Fig.12 for further explanation. More details see in Appendix N.
>
>
> **R 3.4  the experimental results do not confirm whether the improvement is due to network learning better representations or due to improved convergence and shift of the learning curve**
>
> Thanks for your comment. We believe the reported improvements in Table 2 arise from the better learned representations. We could observe that imposing the proposed uniformity metric as an auxiliary penalty
> loss could largely improve its uniformity, although it also lightly damage the alignment, see in Appendix M.  Singular value spectrum in Fig.7 also show the additional loss W_2 benefits the alleviation of the dimensional collapse.
>
>
> **R 3.5  Code is not provided. Given the scope of experiments spanning various techniques it is hard to figure if the results would be reproducible**
>
> For better reproducibility, we provide the code and experiment records in the Supplementary Materials.
>
> As you mention whether the reported improvements are indeed due to the proposed metric or other aspects, we believe the reported improvements arise from the proposed metric indeed, since we keep totally the same experiment setting for (BYOL, BYOL+W_2), (MoCo v2, MoCo v2 + W_2), (BarlowTwins, BarlowTwins + W_2) and (Zero-CL, Zero-CL+W_2).
>
> **R 3.6  The paper was organized and written well with some minor problems**
>
> Thanks for your detailed suggestions. In the revised version, we remove the extra "on the", and move some proofs to the Appendix.
>
> As you mention DINO, we believe it is a really interesting work, and we add the paper to the related work.
>
> As you mention there is no equation 11, we place this equation in the appendix in the previous version, and now we place it in Sec.3.3.
>
> We have made the above statements clearer in the revised version, thanks for your constructive comments.

---

### Official Review · Reviewer_UX2Z · 2022-10-24

**Confidence:** 3
**Correctness:** 3
**Technical Novelty And Significance:** 2
**Empirical Novelty And Significance:** 2
**Recommendation:** 5

**Clarity, Quality, Novelty And Reproducibility:**

The work is well-written and well presented.
There are many existing work studying the representation quality of self-supervised learning, the result and the improvement seem to be a bit incremental.
The reproducibility can be improved if code can be released.



**Strength And Weaknesses:**

# strength: the work studies an interesting problem in self-supervised learning.
The work proposed a new metric that better evaluates the quality of representations learned via self-supervised learning.

# weakness: the reviewer finds the overall result a bit incremental, compared to the work Wang & Isola 2020
1. The theorems in Section 3.1 & Section 3.2 are some known facts,
2. more validation is needed on the assumption that a uniform distribution over the sphere is approximately the same as Gaussian distribution in high dimension
3. the improvement of accuracy on downstream tasks in Table 2 is quite marginal.



**Summary Of The Paper:**

This work presents a systematical analysis of the collapse degree of representations in self-supervised learning. To achieve a quantifiable analysis of the dimensional collapse, the work proposed to use the Wasserstein distance between the distribution of learned representations and the ideal distribution as the metric of uniformity. Based upon this, it introduces five desirable constraints for ideal uniformity metrics to make theoretical comparisons between the proposed metric and the existing one. Finally, the work shows that by imposing the proposed uniformity metric as an auxiliary loss term for various existing self-supervised methods, it consistently improves the downstream performance.



**Summary Of The Review:**

Overall, this is a well-presented paper. However, the reviewer finds the result a bit incremental overall.

---

> ### Author Response · Authors · 2022-11-14
> **To Reviewer UX2Z**
>
> **R 2.1 more validation is needed on the assumption that a uniform distribution over the sphere is approximately the same as Gaussian distribution in high dimension**
>
> As suggested by this comment, we made a lot of efforts during the rebuttal  phase to formally validate the Gaussian assumption in both theoretical and empirical aspects.
>
> - we provide theoretical evidence to prove that $Y_i$ approximates a Gaussian distribution $\mathcal{N}(0, \frac{1}{m})$ when $m$ is large enough, see in Theorem 2 in Sec.3.2.1; In detail, we first seek the probability density function (pdf) of $Y_i$ as shown in App.C. Since the probability density functions of both distributions are known, we could derive the  Kullback-Leibler divergence between them. One trick is to expand a logarithm term using Taylor expansion. Finally,  we obtain that the divergence has a limit of zero when $m$ approaches infinity. See App.D for detailed proof.
> - we provide empirical efforts to check how the closeness between $Y$ and a Gaussian distribution, see in Sec.3.2.2.
>
> As for empirical validation, we add more analysis to support the Gaussian assumption. (i) See the 1D visualization in Fig.2, we analyze the density of $l_2$ normalized Gaussian distribution, and Gaussian distribution over various dimensions, the difference between the two distributions becomes negligible when $m>32$. (ii) we use Wasserstein distance to further quantitatively measure the closeness between two distributions, as visualized in Fig.3, the distance is converged to zero with the large $m$. (iii) We also provide 2D visualization in Fig.8, which also shows the density of the two distributions are close.
>
> **R 2.2 overall result a bit incremental, compared to the work (Wang & Isola 2020)**
>
> The significant difference between the uniformity metric in Wang & Isola 2020 and our uniformity metric, is **whether it could sufficiently deal with dimensional collapse**.
>
> - We **theoretically** demonstrate the baseline metric is insensitive to the dimensional collapse, which our proposed metric could capture sensitivity to the dimensional collapse. See Sec. 4.2  for **theoretical comparison** where we prove our metric could satisfy all these five necessary Desterata while the uniformity metric in Wang & Isola 2020 only satisfies two of them.
>
> - In **empirical comparison** ,  (i) Our proposed uniformity metric is capable of capturing salient sensitivity w.r.t different dimensional collapse degrees, as visualized in Fig.5(a) and Fig.5(b). While the baseline uniformity metric in  Wang & Isola 2020 keeps almost identical when the collapse degree changes from  $0$\% to $80$\%; (ii) we also conduct dimensional collapse analysis w.r.t various dimensions, as shown in Fig.6(a) and Fig.6(b), the baseline uniformity metric in Wang & Isola 2020 fails to identify the dimensional collapse with a large dimension $m \geq 256$, while our proposed uniformity metric can identify the dimensional collapse no matter how great/small $m$ is.
>
>
> **R 2.3 The theorems in Section 3.1 & Section 3.2 are some known facts**
>
> We largely shorten the content regarding  Theorem 1 (include statements and proofs) in the main text, see more concise Sec 3.1. By doing so, we could leverage more space to highlight the core contributions of this work.
>
> In Sec.3.2 we have moved some obvious statements and theorems to the appendix. Moreover, we provide more theoretical results and detailed empirical results on the closeness between $Y$ and a Gaussian distribution, which we believe this part is essential to serve as the foundation of our method.
>
> **R 2.4 The reproducibility can be improved if code can be released.**
>
> For better reproducibility, we provide the code and experiment records in the Supplementary Materials.
>
> We have made the above statements clearer in the revised version, thanks for your constructive comments.

---

### Official Review · Reviewer_9DpF · 2022-10-24

**Confidence:** 4
**Correctness:** 2
**Technical Novelty And Significance:** 2
**Empirical Novelty And Significance:** 2
**Recommendation:** 5

**Clarity, Quality, Novelty And Reproducibility:**

I think that this paper is clearly written. Analyzing uniformity metrics from the listed five measures is nice. However, I doubt there is any genuine theoretical contribution from this paper, despite its lengthy argument. The empirical justification is also quite weak and incomplete in my opinions.

**Strength And Weaknesses:**

Here is a summary of strengths and weaknesses.

Pros:
- The perspective of alignment and uniformity is very helpful for understanding data representation in self-supervised learning, and yet this perspective is not fully explored and some issues such that dimensional collapse remains. The authors suggest that one reason may be that common uniformity metrics do not satisfy Cloning Constraint and feature Baby Constraint properties. This argument seems plausible and provides a new way of thinking uniformity.

Cons:
- Section 3 seems to be very obvious to many people. So it is not clear what the theoretical contribution of this paper is.
- The empirical contribution is also fairly weak: the results in Table 2 are not overwhelmingly convincing, there is no visualization of data representation, there is a lack of in-depth analysis and comparison of different uniformity metrics, etc.
- Estimation of covariance matrices from data, as shown in Eq. 3, is not well grounded. It is well-known in statistics that estimating covariance matrices in high dimensions is not reliable. Also, the Gaussian assumption may not be valid in the representation space.

**Update:** After the revision, I feel that the empirical part is strengthened, so I slightly increased the score.

**Summary Of The Paper:**

In this paper, the authors are motivated by a recent observation in self-supervised learning---in order to prevent a model from dimensional collapse and subsequently poor data representation, we should encourage the representation to be uniform; see for example [1]. The authors argue that zero-mean isotropic Gaussian distribution has the ideal uniformity property, and based on this, they propose to use Wasserstein distance as a uniformity metric as a part of the loss function. It is shown that the Wasserstein distance is useful for avoiding dimensional collapse and improving representation.

[1] Tongzhou Wang and Phillip Isola. Understanding contrastive representation learning through alignment and uniformity on the hypersphere. In ICML, 2020.


**Summary Of The Review:**

The authors study uniformity in self-supervised learning and propose Wasserstein distance as a part of the loss function. I find the overall argument plausible. However, the major weaknesses---namely claiming obvious results as theoretical contributions, and weak empirical justications---severely lower the quality of this paper. Thus, I would not recommend acceptance.

---

> ### Author Response · Authors · 2022-11-14
> **To Reviewer 9DpF (Part 1)**
>
> **R 1.1 Section 3 seems to be very obvious to many people. So it is not clear what the theoretical contribution of this paper is.**
>
> We apologize that the theoretical part of the previous version is relatively weaker since 1) we have not found the theoretical connection between the $l_2$ normalized Gaussian distribution and Gaussian distribution; 2) have not made it clear regarding the different behaviors of the uniformity metrics on dimensional collapse.  As suggested by constructive reviewers, we have fixed the above main issues and we believe that it has strengthened the theoretical part of this paper.
>
> In the revised version, our theoretical contributions are clearer, and can be summarized into two parts: (i) we theoretically demonstrate the baseline uniformity metric in [1] is insensitive to the dimensional collapse, see Claim 2 in Sec.4.2, and demonstrate our proposed uniformity metric could capture salient sensitivity to dimensional collapse, see in Claim 1 in Sec.4.2; (ii) we also provide theoretical evidence to prove that $Y_i$ approximates a Gaussian distribution $\mathcal{N}(0, \frac{1}{m})$ when  $m$ is large enough, see in Theorem 2.
>
> As you may notice the theoretical contribution (i), we provide theoretical evidence about *why the baseline uniformity metric in [1] is insensitive to the dimensional collapse, while our proposed uniformity metric could capture salient sensitivity to that*. To explain this question, we provide more discussions like below.
>
> - Property 5, also named Feature Baby Constraint, can identify the capacity of uniformity metrics on capturing sensitivity to the dimensional collapse.
> The reasonability see in Appendix N.
> - We theoretically demonstrate our proposed uniformity metric could satisfy Property 5, while the baseline uniformity metric in [1] violates Property 5. Detailed proof see in App E.1 and E.2.
> - We also provide empirical evidence to show how two uniformity metrics are sensitive to dimensional collapse. We design representations with different dimensional collapse degrees, by concatenating the zero vectors (i.e., they are full dimensional collapse)  with  sampled data vectors from the standard Gaussian distribution (i.e., ideal uniformity without collapse). See in Fig.5(a) and Fig.5(b), the baseline uniformity metric in [1] keeps almost no change from $0$\% to $80$\% collapse level, while our proposed uniformity metric is sensitive.
>
> As for theoretical contribution (ii), we first seek the probability density function (pdf) of $Y_i$ as shown in App.C. Since the probability density functions of both distributions are known, we could derive the  Kullback-Leibler divergence between them. One trick is to expand a logarithm term using Taylor expansion. Finally,  we obtain that the divergence has a limit of zero when $m$ approaches infinity. See App.D for detailed proof.
>
> [1] Tongzhou Wang and Phillip Isola. Understanding contrastive representation learning through alignment and uniformity on the hypersphere. In ICML, 2020

---

> ### Author Response · Authors · 2022-11-14
> **To Reviewer 9DpF (Part 2)**
>
> **R 1.2 The empirical contribution is also fairly weak: the results in Table 2 are not overwhelmingly convincing, there is no visualization of data representation, there is a lack of in-depth analysis and comparison of different uniformity metrics, etc.**
>
> We have revised the paper about our empirical contributions. This can be summarized into three parts: (i) we provide more empirical analysis to check how the closeness between $Y$ and a Gaussian distribution, see in Sec.3.2.2; (ii) we provide an empirical comparison between two uniformity metrics, see in Sec.4.3; (iii) we impose the proposed uniformity metric as an auxiliary loss term for various existing self-supervised methods, and conduct an in-depth analysis of learned representations including dimensional collapse analysis, convergence analysis on alignment and uniformity metrics, see in Sec.5.
>
>
> For **visualization of data representation**,  we have provided more visualization. See the 1D visualization in Fig.2, the difference between $l_2$ normalized Gaussian distribution and Gaussian distribution becomes negligible when $m>32$. 2D visualization in Fig.8 also shows the density of the two distributions is close. To further quantitatively measure the closeness, we use Wasserstein distance to calculate the distance between two distributions, as visualized in Fig.3.
>
>
> For **in-depth analysis**, our revisions are like below
>
> - we have provided more visualization, see Fig. 2, 3 and 8.
> - we have moved the analysis of the dimensional collapse from the appendix to the main text, which conducts collapse analysis with respect to the different levels of dimensional collapse, sec Sec 4.3.
> - following [2], we have checked the singular values of the covariance matrix of representations in downstream tasks (e.g., CIFAR-100 dataset). The results (see  Fig.7) shows imposing our proposed uniformity could largely reduce dimensional collapse, see Sec 5.
>
>
>
> For **comparison of different uniformity metrics**,
> The comparison of different uniformity metrics is two-fold
>
> - please check Sec. 4.2 for **theoretical comparison** between these two uniformity metrics (we have additionally provided a proof on Theorem 2).
> - please check Sec. 4.3 for **empirical comparison** between these two uniformity metrics using synthetic data. Our comparisons can be summarized as two parts: (i) We first design representations with different dimensional collapse degrees, as visualized in Fig.5(a) and Fig.5(b), our proposed uniformity metric is capable of capturing salient sensitivity, while the baseline uniformity metric in [1] keeps almost no change from  $0$\% to $80$\% collapse level; (ii) we also conduct dimensional collapse analysis w.r.t various dimensions, as shown in Fig.6(a) and Fig.6(b), the baseline uniformity metric in [1] fails to identify the dimensional collapse with a large dimension $m \geq 256$, while our proposed uniformity metric is able to identify the dimensional collapse no matter how great/small $m$ is.
>
> In summary,  the existing uniformity metric cannot distinguish the dimensional collapse while the proposed uniformity metric could. (i) Theoretically, the proposed uniformity metric can satisfy Property 4 and 5 while the existing one cannot, which are related to dimensional collapse, see Sec. 4.2; (ii) the proposed uniformity metric is more distinguishable than the existing one with respect to different collapse levels and dimension sizes, sec Sec.4.3.
>
> For better **reproducibility**, we provide the code, also with experiment records in the Supplementary Materials.
>
> [1] Tongzhou Wang and Phillip Isola. Understanding contrastive representation learning through alignment and uniformity on the hypersphere. In ICML, 2020
>
> [2] Li Jing, Pascal Vincent, Yann LeCun, and Yuandong Tian. Understanding dimensional collapse in contrastive self-supervised learning. In ICLR 2022
>
>
> **R 1.3 Estimation of covariance matrices from data, as shown in Eq. 3, is not well grounded. It is well-known in statistics that estimating covariance matrices in high dimensions is not reliable. Also, the Gaussian assumption may not be valid in the representation space**
>
> Thank the reviewer for pointing out that estimating covariance matrices in high dimensions is not reliable. But it is a common practice in many works, such as in [3], [4], [5].  We leave this part as future work.
>
> [3] Adrien Bardes, Jean Ponce, Yann LeCun. VICReg: Variance-Invariance-Covariance Regularization for Self-Supervised Learning. in ICLR 2022.
>
> [4] Li Jing, Pascal Vincent, Yann LeCun, Yuandong Tian. Understanding Dimensional Collapse in Contrastive Self-supervised Learning. in ICLR 2022.
>
> [5] Jianlin Su, Jiarun Cao, Weijie Liu, Yangyiwen Ou. Sentence Representations for Better Semantics and Faster Retrieval.
>
>
> We have made the above statements clearer in the revised version, thanks for your constructive comments.

---

### Author Response · Authors · 2022-11-14
**To All Reviewers**

Thanks to all reviewers for their constructive comments. Especially, thank reviewers *UX2Z* and  *ZorS*  for recognizing that the paper is clearly written.
Thank *iVeQ* for stating the proposed metric is very simple to implement for uniformity.
Thank the reviewer  *9DpF* for commenting that might provide a new way of thinking uniformity (one of the core concepts in self-supervised learning).

We thoughtfully revise our papers according to the reviewers' suggestions.
First, we replace the factly-ungrounded statements with concrete theoretical conclusions on the following points:

- we  provide and prove a new theorem ( Theorem 2) statest that $Y_i$ approximates a Gaussian distribution $\mathcal{N}(0, \frac{1}{m})$ when  $m$ is large enough. This would make our assumption of the new uniformity metric more solid.

- we theoretically demonstrate the existing uniformity metric in Wang & Isola (2020)  is insensitive to dimensional collapse, see Claim 2 in Sec.4.2.

Second, we conduct more empirical analysis, and our empirical contributions can be summarized into three parts:

- we provide empirical analysis to check how the closeness between $Y$ and a Gaussian distribution, including 1D visualization, 2D visualization, and quantitative analysis on closeness by Wasserstein distance, see Sec.3.2.2.

- we provide an empirical comparison between two uniformity metrics, see Sec.4.3.

- we impose the proposed uniformity metric as an auxiliary loss term for various existing self-supervised methods, and conduct an in-depth analysis of learned representations including dimensional collapse analysis, convergence analysis on alignment, and uniformity metrics, see Sec.5.

Third, we largely shorten the content regarding  Theorem 1 (including statements and proofs) in the main text, which is believed by some reviewers to be obvious. By doing so, we could leverage more space to highlight the core contributions of this work.


We release our codes for reproducibility in the Supplementary Material.

---

### Decision · Program_Chairs · 2023-01-20

**Decision:**

Reject

**Justification For Why Not Higher Score:**

As pointed out by reviewers, some of the results in section 3 are known, and it needs to improve the section. Moreover, it needs more experimental results to justify the claim (as pointed out by reviewers).

**Justification For Why Not Lower Score:**

N/A

**Metareview: Summary, Strengths And Weaknesses:**

In this paper, the authors propose a new self-supervised representation learning that handle the uniformity during the learning. More specifically,  the authors propose the Wasserstein distance between an isotropic zero-mean Gaussian distribution and the distribution of the learned representations as a robust uniformity metric for self-supervised learning. Through experiments, the authors demonstrate that the proposed method helps improving the classification accuracy. Using the Wasserstein distance is reasonable. However, as pointed out by reviewers, some of the results in section 3 are known, and it needs to improve the section. Moreover, it needs more experimental results to justify the claim (as pointed out by reviewers). Thus, I encourage authors to revise the paper based on the reviewer's comments and resubmit it to a future venue.

**Summary Of Ac-Reviewer Meeting:**

N/A